# Sailing in High-Dimensional Spaces: Low-Dimensional Embeddings through Angle Preservation

## Abstract

Low-dimensional embeddings (LDEs) of high-dimensional data are ubiquitous in science and engineering. They allow us to quickly understand the main properties of the data, identify outliers and processing errors, and inform the next steps of data analysis. As such, LDEs have to be *faithful* to the original high-dimensional data, i.e., they should represent the relationships that are encoded in the data, both at a local as well as global scale. The current generation of LDE approaches focus on reconstructing *local distances* between any pair of samples correctly, often outperforming traditional approaches aiming at all distances. For these approaches, global relationships are, however, usually strongly distorted, often argued to be an inherent trade-off between local and global structure learning for embeddings. We suggest a new perspective on LDE learning, reconstructing *angles* between data points. We show that this approach, Mercat, yields good reconstruction across a diverse set of experiments and metrics, and preserve structures well across all scales, outperforming existing methods across datasets and metrics in most cases by a margin. Compared to existing work, our approach also has a *simple formulation*, facilitating future theoretical analysis and algorithmic improvements.

## 1 Introduction

A key aspect of modern data analysis is data visualization. Usually employed early in an analysis, such a visualization can help to identify errors in data recording and preprocessing, discover outliers, and overall structure of the data, informing next processing steps or choice of further analysis. Low-dimensional embeddings (LDEs) methods take on this task, computing a 2- or 3-dimensional embedding of the data preserving some essential structures. Such methods are nowadays widely used in e.g. biology for interpreting complex gene regulation (Kobak & Berens, 2019), or in explainable machine learning to investigate latent spaces of neural networks (Li et al., 2021; Zhang et al., 2021; Rostami et al., 2023).

To get a proper understanding of the data and then make informed decisions, the LDEs have to be faithful to the original data: *local structures* should be perceivable, but also *global relationships* of these structures should be appropriately reflected. While several widely used methods to obtain LDEs exist, it has been observed that often they only reconstruct local structures faithfully, while neglecting global structures (Moon et al., 2019; Wang et al., 2021; Ma et al., 2023; Chari & Pachter, 2023; Sun et al., 2023). This leads to a loss of information in the embedding space, where for data consisting of clusters most inter-cluster information is lost (Cai & Ma, 2022), and for data with manifold structures, the manifold gets absurdly distorted or torn, capturing only locally faithful information that make it hard to reason about the data as a whole (Kobak & Linderman, 2021b; Meilă & Zhang, 2023; Xia et al., 2024) (cf. Fig. 2).

This drawback likely comes by design, as current state-of-the-art approaches mainly focus on the correct *reconstruction of local distances*, neglecting long-range distances in the process. A common argument for this approach is that it is impossible to compress all information in the high-dimensional data into the low-dimensional space. A theoretical barrier is expected to exist which reflects the fundamental trade-off between local and global structure preservation inherent in LDEs. Due to the iterative nature of the optimization and the rather complex objective functions involved

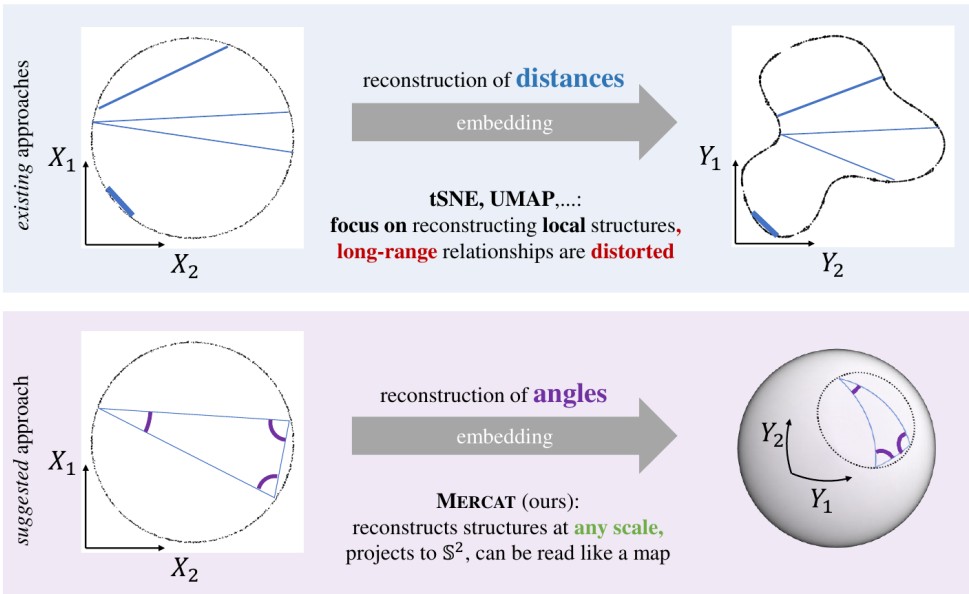

Figure 1: *Visual abstract.* Existing work (top) optimizes low-dimensional embeddings to *reconstruct distances*, focusing on reconstruction of local structures (smaller distances), leading to distortion or breaking of global structures (larger distances). We suggest (bottom) to *reconstruct angles* between any three points, embedding on the sphere 2D sphere $\mathbb{S}^2$, capturing structures at any scale.

in most of the existing approaches, theoretical understanding is still limited (Linderman & Steinerberger, 2019; Cai & Ma, 2022; Damrich & Hamprecht, 2021). Nevertheless, empirical studies (Böhm et al., 2022; Wang et al., 2021) suggest it is unlikely that existing methods have already reached such a critical point, that any further improvement on the global structure preservation has to come with a compromise in the faithfulness of local structure preservation.

In this work, we consider a new approach that is distinct from the common paradigm of reconstructing (local) distances and propose a simple new LDE approach preserving both local and global structures well. Inspired by a breakthrough in navigation in the 16th century, the Mercator Projection, we aim for a representation that focuses on the *reconstruction of angles*. The motivation for the Mercator Projection—a projection of planet Earth on a 2D map—was that flat lines on the map represent routes of constant bearing, greatly simplifying navigation with a compass. This 2D map, which most of us recognize from world maps in an atlas or online mapping services, implements the idea of preserving angles locally at every point, thus keeping relative orientation of objects (landmasses) intact. We follow this idea and introduce the concept of a *angle-approximating embeddings* (Fig. 1). Similar in spirit to a Mercator projection we keep the overall arrangement of objects (here, data points) by modeling the orientation of objects to each other (angles between data points), but not only consider local angles, but also global angles to correctly reconstruct the data. We define as a lower dimensional representation of the original data in which angles between triplets of data points are preserved, mapping a high-dimensional dataset on the 2D unit sphere $\mathbb{S}^2$, which can be directly visualized.

Our approach, MERCAT, is both theoretically appealing due to its simplicity, and practically useful. We show on challenging toy examples, synthetic data studies, and real data sets, that the embeddings are not only visually more faithful to the original data but also quantitatively great in reconstruction of both local and global structure. It, hence, serves as a basis for new developments in LDE theory and practice. Concretely, our **contributions** are (i) we **propose a new paradigm** for computing low-dimensional embedding by optimizing for reconstruction of angles rather than distances, (ii) provide **efficient algorithmic ideas** to compute such an LDE in practice, (iii) give **empirical and theoretical justifications** for our algorithmic ideas, and (iv) provide **extensive evaluation** on synthetic and real-world data against state-of-the-art approaches with a diverse set of metrics.

## 2 Related Work

Perhaps the most widely known dimensionality reduction method is principal component analysis (PCA) (Pearson, 1901), followed by seminal work on multidimensional scaling (Torgerson, 1952), self-organizing maps (Kohonen, 1982), and Laplacian eigenmaps (Belkin & Niyogi, 2001). For all of these methods, their objective usually focuses on getting the *larger* distances right.

**Local reconstruction.** With empirical evidence and the methodological insight that data often lies on an intrinsically low-dimensional manifold, subsequent work such as local linear embedding (Roweis & Saul, 2000), Isomap (Tenenbaum et al., 2000), and Hessian eigenmaps (Donoho & Grimes, 2003) focus on getting *local* distances right and modeling relationships *non-linearly*. However, these methods rely on stringent manifold assumptions, restricting their scalability for large and high-dimensional data, and making their practical performance susceptible to noise and data outliers. More recently, a family of low-dimensional embedding algorithms based on ideas of stochastic neighbor embeddings (SNE) (Hinton & Roweis, 2003), with t-distributed SNE (tSNE) (van der Maaten & Hinton, 2008) and the closely related Uniform Manifold Approximation (UMAP) (McInnes et al., 2018) being its most prominent representatives, have become extremely popular in data analysis and scientific research, especially in the field of molecular biology (Kobak & Berens, 2019; Kobak & Linderman, 2021b). These algorithms again focus on the reconstruction of local neighborhoods, but have been found more scalable and more robust to high-dimensional noisy data sets, compared with previous methods.

**Studied limitations.** While widely employed, both tSNE and UMAP as well as related approaches (Linderman et al., 2019; Artemenkov & Panov, 2020) suffer from several known issues, one of them being that densities are not properly preserved in the embeddings – two differently sized clusters are mapped to the same amount of space in the embedding. Another limitation is the severe distortions of global structures caused by the neglect of long-range distances in the reconstruction (Chari & Pachter, 2023; Kozlov, 2024; Lause et al., 2024). Various metrics have been developed to assess embedding distortions (Venna et al., 2010; Xia et al., 2024). Recent works, such as densmap, focused on solving these issues (Narayan et al., 2021; Fischer et al., 2023). Interestingly, early work on Isomap extended the original method normalizing distances for a local point by the local density around that point, thus similarly reflecting local densities of the data in the objective (de Silva & Tenenbaum, 2002). The authors showed that this approximates conformality in the mathematical sense, thus coining it C-Isomap.

**Hyperbolic embeddings** An orthogonal line of work considers embedding data into hyperbolic spaces (Walter, 2004; Nickel & Kiela, 2017; Bläsius et al., 2018). These methods are particularly suited for capturing hierarchical structures by leveraging the intrinsic geometry (shape) of a hyperbolic space, yet less ideal for general manifolds or clustered data. Unlike our work that focuses on angle preservation, these methods still aim to reconstruct distances but in the hyperbolic space.

## 3 Angle-preserving low-dimensional embeddings

Our key idea is motivated by the central issue of state-of-the-art LDE approaches, which is the poor reconstruction of long-range, or "global", relationships, such as the orientation of sub-manifolds or the relative locations of clusters (Fig. 2). This problem comes by design, as concurrent work, e.g., tSNE and UMAP, focuses on reconstructing local distances well. This approach so far outperforms traditional methods aiming to reconstruct *all* distances, revealing more meaningful structure. We, however, instead hypothesize that the problem is inherent in the modeling of distances. Inspired by the Mercator projection, which revolutionized navigation in the 16th century by providing an locally angle-preserving (conformal) 2D map of the earth, we suggest to compute an **embedding that approximately reconstructs angles between any three data points** (angles within each triangle, see Fig. 1). As opposed to classical conformal maps, we further aim to reconstruct *all* angles, including those between distant points, to recover orientation of objects at different scales. As such, it inherently balances global and local relationships by being independent of the scale of the structures (we refer to Sec. 3.2 for a detailed discussion).

Following the idea of a map similar to a Mercator projection, which is a map of fixed size and is without borders (i.e., "leaving" the map on the left means "entering" it on the right), we compute an embedding on the unit sphere. The unit sphere is a 2-dimensional space of fixed area and without

borders, which allows to embed complex and possibly periodic patterns commonly arising from biological applications such as cell linage (Wagner & Klein, 2020), cell cycle (Liang et al., 2020; Saelens et al., 2019) and circadian rhythm (Auerbach et al., 2022), but at the same time lends itself for efficient computation of angles and (geodesic) distances.

### 3.1 FORMAL DESCRIPTION

For data $X = \{X_i\}_{1 \le i \le n} \subset \mathbb{R}^d$ of $n$ samples and $d$ features, we are interested in a good reconstruction of $X$ in a low-dimensional space, particularly the (unit) 2-sphere $\mathbb{S}^2$. As introduced above, we consider an approximation of a conformal embedding for our LDE, i.e. an embedding that *approximately reconstructs angles* from the data in the high-dimensional space, thus orienting both local as well as global structures properly to each other. More formally, we are searching for a map $X_i \mapsto Y_i$, where $Y_i \subset \mathbb{S}^2$, such that for any sample $i$ and pair of samples $j, k$, the (Euclidean) angle between $X_j$ and $X_k$ measured at $X_i$ should be reconstructed in $Y = \{Y_i\}_{1 \le i \le n}$, i.e., $\angle X_j X_i X_k \approx \angle Y_j Y_i Y_k$. Here $\angle Y_j Y_i Y_k$ is the angle between the shortest paths $\overline{Y_i Y_j}$ and $\overline{Y_i Y_k}$ *on the $\mathbb{S}^2$ sphere*, i.e., the angle between the corresponding geodesics. Using the definition of Euclidean inner product in $\mathbb{R}^d$, we get $\arccos\left(\frac{(X_j - X_i)^\top (X_k - X_i)}{||X_j - X_i|| \; ||X_k - X_i||}\right) \approx \angle Y_j Y_i Y_k$. We parameterize each point $Y_i$ in $\mathbb{S}^2$ by two parameters $(\phi_i, \theta_i)$, which correspond to longitude and latitude on the 2-sphere. The angles can then be computed in terms of these coordinates using the well-known geometric relations on sphere (Appendix A.1).

**Objective.** Considering the root mean square deviation of angles in $Y$ from those corresponding in $X$, we get a differentiable objective

$$\mathcal{L}(X, Y) = \left(\frac{1}{n} \sum_i \frac{1}{(n^2 - n)/2} \sum_{(i,j,k): k > j \text{ and } j,k \ne i} ||\angle X_j X_i X_k - \angle Y_j Y_i Y_k||_2^2\right)^{1/2}, \quad (1)$$

which we can optimize as $\arg \min_{Y \subset \mathbb{S}^2} \mathcal{L}(X, Y)$. Through the parameterization by longitude and latitude on the 2-sphere as defined above, we can optimize directly on the sphere by standard gradient descent on $\frac{\partial \mathcal{L}(X,Y)}{\partial \phi}, \frac{\partial \mathcal{L}(X,Y)}{\partial \theta}$. Having all components of our approach together, we give the pseudocode of our method—named as MERCAT in reminiscence of the inspirational idea from the Mercator projection—in Algorithm 1.

---

**Algorithm 1** MERCAT

**Require:** $X \in \mathbb{R}^{n \times d}$ input data; $r$ dimension for spectral denoising; $i_{\max}$ number of iterations; $l$ learning rate; $L$ learning rate schedule
**Ensure:** $Y$ as low-dimensional embedding of $X$ on $\mathbb{S}^2$
  $\hat{X} \leftarrow PCA(X)_{1:r}$                        ▷ PCA reduction for robustness, see Sec. 3.3,3.4
  $Y \leftarrow [PCA(X)_1, PCA(X)_2]$         ▷ initialization: wrap first two PCs around half sphere
  $Y \leftarrow Y - \min(Y), Y \leftarrow 0.6\pi \frac{Y}{\max(Y)} + 0.2\pi$      ▷ geodesic coords, push away from poles
  **for** $i \in \{1, ..., i_{\max}\}$ **do**
      Compute $\mathcal{L}(\hat{X}, Y)$             ▷ see Eq. 1 and subsampling consideration in Sec. 3.3
      Update $Y$ w.r.t. $\frac{\partial \mathcal{L}(\hat{X}, Y)}{\partial Y}$          ▷ use Adam (Kingma & Ba, 2015) for gradient updates
      Update $l$ w.r.t. $L$
  **end for**
  **return** Y

---

### 3.2 "SCALE-INVARIANCE" OF ANGLE PRESERVATION

We want to emphasize that angle preservation is in some sense invariant to scale, which is why this approach recovers both local as well as global structure almost equally well, which we believe to be a key advantage of angles over distances. To see this difference between the approaches, consider the example of three points $A$, $B$, $C$. Scale the distance $AB$ and $AC$ both by a constant $c$ while keeping the angle intact. In reconstruction, the angle at $A$ will receive the same importance during optimization regardless of scaling factor $c$. This is in stark contrast to distance-based reconstruction

methods, for methods like MDS, the "weight" of a pair of points in the optimization is increased the further away they are — it would be sensitive to $c$. Similarly, for neighborhood-based methods such as tSNE and UMAP, assuming any other points stay at the same location in the above example, the points $B$ and $C$ lose relevance for the location of $A$ the further they are away from $A$ (the larger the constant $c$). Hence, where traditional (distance-based) methods do have a preference to the relative location of points to a target point (the scale of how far points are away), our (angle-based) approach is in some sense invariant to that scale.

### 3.3 COMPUTATIONAL AND STATISTICAL STRATEGIES

**Initialization.** Low-dimensional embedding techniques greatly benefit from a good initialization (Kobak & Linderman, 2021a). For a good initial embedding we follow the established strategy and consider the first two principal components as initialization, wrapping them around the sphere. In particular, let $PC1$ and $PC2$ be the points in $X$ projected onto the first and second principal component, respectively. We then compute the initial longitudes as $0.6\pi \frac{PC1 - \min(PC1)}{\max(PC1) - \min(PC1)} + 0.2\pi$ and initial latitudes as $0.6\pi \frac{PC2 - \min(PC2)}{\max(PC2) - \min(PC2)} + 0.2\pi$. We thus roughly distribute the points on a half-sphere while keeping the initial estimates away from the poles[1]. This yields an initial ordering of points relative to each other, yet MERCAT significantly refines this initial embedding. We provide snapshots during the optimization along with the initial embedding in App. Fig. 7.

**Two simplifying computational tricks.** In practice, we can employ two computational tricks to accelerate the optimization and improve numerical stability. First, we drop the arc-cosine from angle computations, i.e., we compute differences between normalized dot products, which is a *strictly monotone transformation* of the original formulation that does not change the optima. Second, we compute angles on $Y$ by pure linear algebra, using the dot product between normals of two planes (see App. Sec. A.2), which can be much faster when employed on modern hardware such as GPUs. We further incorporate two statistical techniques, which speed up the algorithm and improve its scalability while making the embeddings robust to noise and high dimensionality.

**Angle evaluation after spectral denoising.** The first statistical strategy is to denoise the original high-dimensional data using spectral methods before evaluating their Euclidean angles. It is known that in high dimensions, the Euclidean angles between data points can be sensitive to noise perturbations and may suffer severely from the effect of high dimensionality. We argue that in many applications the observed high-dimensional data points are only noisy versions of some latent noiseless samples incorporating certain low-dimensional signal structures. As such, the quantity of interest should be the Euclidean angles among the noiseless samples, of which the angles among the original noisy high-dimensional data can be very poor estimates (see Fan & Zhou (2016); Fan et al. (2018); Fan & Jiang (2019) and Theorem 2 below). To overcome such limitations, instead of directly calculating the angles among the original high-dimensional data points, we propose to first apply a principal component analysis (PCA) to the data matrix $X$, to obtain denoised low-dimensional spectral embeddings given by the leading $r$ principal components. After that, we use the Euclidean angles calculated from such spectral embeddings to estimate the angles among the noiseless samples. Note that while concurrent work on low-dimensional embeddings often considers a similar approach of projecting high-dimensional data to a few principal components before embedding them, they lack theoretical guarantee with respect to their final objective. We here provide rigorous theoretical analysis of our denoising procedure on angle preservation in Section 3.4.

**Subsampling.** In practice, computing all angles in every iteration would incur a computational cost in $O(kn^3)$ for $k$ iterations and $n$ datapoints. While much of it can be efficiently computed by using linear algebra instead of trigonometry, it is still hard to scale to large datasets. We thus investigate whether it is indeed necessary to compute *all* angles in every iteration. For an empirical study, we consider a real dataset about single-cell gene expression of human hematopoiesis (Paul et al., 2015), a typical application for low-dimensional embeddings. We sample $n = 500$ points and compute for each point $X_i$ all angles at that point, i.e., all $\angle X_j X_i X_k, j \neq k \neq i$, yielding matrices of cosine-angles $\Theta_i[j,k] = \cos(\angle X_j X_i X_k)$. We then compute a singular value decomposition for each of these matrices (cf App. Fig. 4a), which show that the matrices have only few large singular values. Furthermore, computing the effective rank of the matrix (Roy & Vetterli, 2007), which give an estimate of the intrinsic dimensionality of the matrix based on the singular values, we observe

---

[1]Having points close to a pole leads to slow optimization as the loss landscape is flat around them.

that the effective rank is very low (cf App. Fig.4b), with a mean of $13.8$. Based on these insights, rather than computing all angles at every point, we suggest to sample a fraction of points at random each time we compute angles at point $X_i$, i.e., we consider $\angle X_j X_i X_k, j \neq k$ and $j, k \in S(n) \setminus \{i\}$, where $S(n)$ is a random subset of $[n]$. For the remainder of the paper, in every iteration for each point $X_i$ we will draw 64 other points uniformly at random and compute angles at $\angle X_{j_1} X_i X_{j_2}$, where $j_1, j_2$ are from these sampled subsets, effectively reducing the computational costs to $\mathcal{O}(kn)$. In App. Fig. 5 we provide an empirical study of how gradients estimated through subsampling differ from gradients estimated using all angles, showing that the cosine similarity between gradients is close to 1, further emphasizing the effectiveness of the subsampling approach.

## 3.4 THEORETICAL JUSTIFICATIONS

Here we provide theoretical justification for the spectral estimation of the true Euclidean angles of the input data. For a theoretical reasoning of the efficacy of angle subsampling, we refer to App. Sec. B.3. A given input data is usually noisy and high-dimensional but contains low-dimensional structures. To fix ideas, we first introduce the statistical framework of the spiked population model (Bai & Ding, 2012; Bai & Yao, 2012; Baik & Silverstein, 2006; Bao et al., 2022; Johnstone, 2001; Paul, 2007). We assume the high-dimensional data matrix $X \in \mathbb{R}^{n \times d}$ satisfies $X = \sum_{i=1}^{r} \sqrt{\lambda_i} \mathbf{u}_i \mathbf{y}_i^\top + Z = U Y^\top + Z$ where $Z \in \mathbb{R}^{n \times d}$ is the noise matrix, $U = [\mathbf{u}_1 \quad ... \quad \mathbf{u}_r] \in \mathbb{R}^{n \times r}$ has orthonormal column vectors being the latent $r$-dimensional factors or sample embeddings characterizing the underlying signal structure among the $n$ samples, and $Y = [\sqrt{\lambda_1} \mathbf{y}_1 \quad ... \quad \sqrt{\lambda_r} \mathbf{y}_r] \in \mathbb{R}^{d \times r}$ contains the feature loadings whose $(i, j)$ entry characterizing the weight of $j$th latent factor $\mathbf{u}_j$ in the $i$th feature. The above model essentially assumes that the data matrix $X$ contains a latent low-rank signal structure, which complies with many real applications and can be empirically verified by comparing the magnitude of the first few singular values with the other singular values. We assume the noise matrix $Z$ and the (rescaled) feature loading vectors $\{\mathbf{y}_i\}_{1 \leq i \leq r}$ contain independent entries with zero mean and unit variance, but also remark that extensions to more general settings is possible (see discussions after Thm 1). Here, unlike classical theory of PCA where samples are assumed to be independent and features are correlated, we exchange the roles of the samples and features and model the underlying low-dimensional structure among samples by the latent factor $U$, or the low-rank correlation structure among the samples.

From the above model, the high-dimensional data matrix $X$ is a noisy realization of the low-dimensional latent signal matrix $U$ that encodes the true relationship among the samples. The true Euclidean angles between the noiseless samples $j$ and $k$ with respect to sample $i$ can be defined as $\theta_{jk,i} = \arccos\left(\frac{(U_j - U_i) \cdot (U_k - U_i)}{\|U_j - U_i\| \|U_k - U_i\|}\right)$, where $U_i \in \mathbb{R}^r$ is the $i$th row of $U$, giving the true embedding of sample $i$. Our goal is to obtain reliable estimators of the latent Euclidean angles $\{\theta_{jk,i}\}$ based on the noisy data $X$. In our algorithm, we use the leading $r$ eigenvectors $\{\widehat{\mathbf{u}}_i\}_{1 \leq i \leq r}$ of the Gram matrix $X X^\top$ (suppose the data is centered), and estimate $\theta_{jk,i}$ by $\widehat{\theta}_{jk,i} = \arccos\left(\frac{(\widehat{U}_j - \widehat{U}_i) \cdot (\widehat{U}_k - \widehat{U}_i)}{\|\widehat{U}_j - \widehat{U}_i\| \|\widehat{U}_k - \widehat{U}_i\|}\right)$, where $\widehat{U}_i \in \mathbb{R}^r$ is the $i$th row of $\widehat{U} = [\widehat{\mathbf{u}}_1 \quad ... \quad \widehat{\mathbf{u}}_r]$. Our first result concerns the consistency of the latent angle estimation. For any pair $(i, j)$, we obtain the error bound for $|\widehat{U}_i^\top \widehat{U}_j - U_i^\top U_j|$. The accuracy of estimating $U_i^\top U_j$ using $\widehat{U}_i^\top \widehat{U}_j$ is fundamental here since by

$$\frac{(U_j - U_i)^\top (U_k - U_i)}{\|U_j - U_i\| \|U_k - U_i\|} = \frac{U_j^\top U_k - U_i^\top U_k - U_j^\top U_i + \|U_i\|^2}{\sqrt{\|U_j\|^2 + \|U_i\|^2 - 2 U_j^\top U_i} \sqrt{\|U_k\|^2 + \|U_i\|^2 - 2 U_k^\top U_i}} \quad (2)$$

the pairwise inner products $\{U_a^\top U_b : a, b \in \{i, j, k\}\}$ are the building blocks for the angle $\theta_{jk,i}$. In other words, the consistency of $\{U_a^\top U_b : a, b \in \{i, j, k\}\}$ implies the consistency of $\widehat{\theta}_{jk,i}$. Below we obtain the high-probability limit for the estimation error, which guarantees the estimation accuracy of $\widehat{\theta}_{jk,i}$ under sufficiently large signal-to-noise ratio.

To better present our results, we denote the aspect ratio $\phi = \frac{n}{d}$ and assume that $n^{1/C} \leq d \leq n^C$ for some constant $C \geq 1$, characterizing the high dimensionality of the data. We define the rescaled Gram matrix $Q = \frac{1}{\sqrt{dn}} X X^\top$ and denote the population covariance $\Sigma = d^{-1} \mathbb{E}(X X^\top) = I_n + U D U^\top = I_n + \phi^{1/2} \sum_{i=1}^{r} \sigma_i \mathbf{u}_i \mathbf{u}_i^\top$, where $D = \text{diag}(\phi^{1/2} \sigma_1, ..., \phi^{1/2} \sigma_r)$, and $\sigma_1 \geq \sigma_2 \geq ... \geq \sigma_r > 0$, so that $\{1 + \phi^{1/2} \sigma_i\}_{1 \leq i \leq r}$ are the leading $r$ eigenvalues of $\Sigma$.

**Theorem 1** (Guarantee of spectral angle estimators). *Suppose that $\sigma_1 \geq ... \geq \sigma_r \geq 1 + c$ for some constant $c > 0$. Then for any $i, j \in \{1, 2, ..., n\}$, we have that, as $(n, d) \to \infty$,*

$$|\widehat{U}_i^\top \widehat{U}_j - U_i^\top U_j| = \sum_{k=1}^r \frac{(1 + \phi^{1/2}\sigma_k)u_{ki}u_{kj}}{\sigma_k(\sigma_k + \phi^{1/2})} + O_P(n^{-1/2+\epsilon}), \tag{3}$$

*for any small constant $\epsilon > 0$, where we denote $\mathbf{u}_k = (u_{k1}, ..., u_{kn})$, for $1 \leq k \leq r$.*

From the above theorem and Equation (2), we see that the spectral angle estimator $\widehat{\theta}_{jk,i}$ can be arbitrarily close to $\theta_{jk,i}$ as the overall signal strength of the low-dimensional structure, characterized by the parameters $\{\sigma_1, ..., \sigma_r\}$, increases. Our analysis holds for general $\phi$, which may depend on $n$ and needs not to converge in $(0, \infty)$. In particular, our result implies the consistency of $\widehat{\theta}_{jk,i}$ for any low-dimensional structures contained in $U$, that is, for any $\epsilon > 0$, there exist sufficiently large $(\sigma_1, ..., \sigma_r)$ such that

$$\lim_{n\to\infty} P(|\widehat{\theta}_{jk,i} - \theta_{jk.i}| > \epsilon) = 0. \tag{4}$$

We remark that the homoscedasticity assumption on the entries of the noise matrix $Z$ and the feature loading vectors $\{\mathbf{y}_i\}_{1 \leq i \leq r}$ may be relaxed to more general settings, following the universality arguments in random matrix theory (Erdős & Yau, 2017). Moreover, in the above discussion, we took the *normalized shape* captured by the orthonormal latent factors in $U$, as the low-dimensional structure of interest and establish the consistency of $\widehat{U}$. However, we note that in applications where the unnormalized shape is wanted, the eigenvalue-weighted eignvectors may be used for embedding . The consistency of such methods follows from Theorem 1 and the classical eigenvalue perturbation bound (Bhatia, 2007).

Our next result concerns the non-negligible effect of high-dimensionality on the latent angle estimation. Here we assume $\phi$ remains bounded away from zero, that is, $\phi > c$ for some absolute constant $c > 0$. We show that the angles between the original high-dimensional data points, that is, $\bar{\theta}_{jk,i} := \arccos\left(\frac{(X_j - X_i)^\top (X_k - X_i)}{\|X_j - X_i\|\|X_k - X_i\|}\right)$, can be substantially biased with respect to the latent angles.

**Theorem 2** (Limitation of naive angle estimators). *Under the assumption of Theorem 1, if we denote $X_i \in \mathbb{R}^p$ as the ith row of $X \in \mathbb{R}^{n \times d}$, it then holds that, for all $C > 0$, there exist some $\Sigma$ with $\sigma_1, ..., \sigma_r > C$ such that, for any distinct $i, j, k \in \{1, 2, ..., n\}$, $\lim_{n\to\infty} P\left(|\bar{\theta}_{jk,i} - \theta_{jk,i}| \geq \delta\right) = 1$, for some fixed constant $\delta > 0$ that only depends on $\theta_{jk,i}$.*

Comparing this with Eq. 4, we can see that for high-dimensional data, the naive angle estimators $\bar{\theta}_{jk,i}$ based on the original noisy high-dimensional observations can be substantially biased, regardless of signal strength. Theorems 1 and 2 together provide a theoretical justification and explain the practical advantages of our spectral angle estimators for dealing with noisy high-dimensional data.

## 4 EXPERIMENTS

To evaluate MERCAT, we consider low-dimensional manifold datasets exemplifying unsolved issues in current LDEs, synthetic high-dimensional clustered data common in the literature, and real world applications including biology. We compare against the state-of-the-art UMAP (McInnes et al., 2018), TSNE (van der Maaten & Hinton, 2008), DENSMAP (Narayan et al., 2021), and NCVIS (Artemenkov & Panov, 2020), and a fast implementation of hyperbolic MDS HMDS (Keller-Ressel & Nargang, 2020). Due to space constraints we postponed further ablations and results, including the approximately conformal Isomap algorithm C-ISOMAP (de Silva & Tenenbaum, 2002; You & Shung, 2022), UMAP embeddings on the 2D sphere SPHEREMAP (see App. C.1), Laplacian Eigenmaps LAPEIGMAP (Belkin & Niyogi, 2001), Diffusion maps DIFFMAP (Nadler et al., 2005), and non-metric MDS (NMMDS) as a baseline method for global reconstruction to the appendix. Methods are aborted if not delivering a result within 24 hours. We consider a diverse set of metrics that measure how well properties of the high-dimensional data are preserved in the low-dimensional space. In particular (i) *distance preservation ($\|.\|$)*, (ii) *preservation of angles ($\angle$)* measuring how well the angles between data-points are preserved, (iii) *neighborhood preservation ($\therefore$)* measuring how well the closeness of local neighbors is preserved, and (iv) *density preservation ($\odot$)* measuring how well local sample density is preserved. All metrics are in the range $[-1, 1]$, higher is better, and we provide all details and definitions in App. C.2. Whereas

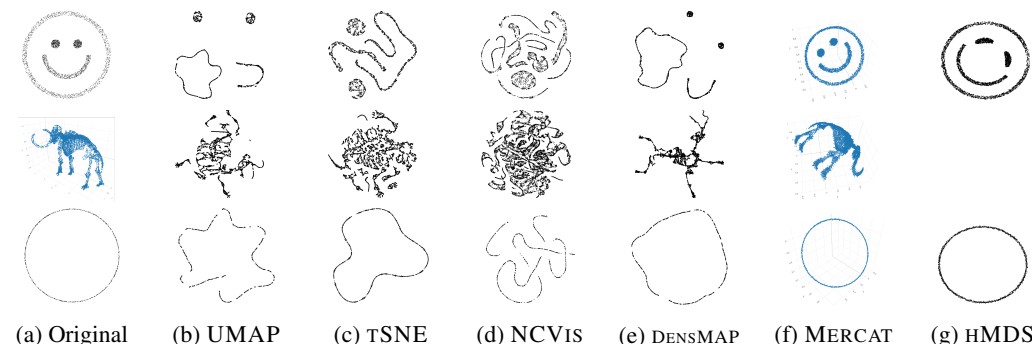

| (a) Original | (b) UMAP | (c) TSNE | (d) NCVIS | (e) DENSMAP | (f) MERCAT | (g) HMDS |

Figure 2: *Embeddings of low-dimensional examples.* We visualize the Smiley (top), Mammoth (middle), and Circle (bottom) data and computed embeddings.

(i) is a standard measure to assess global reconstruction, (ii) is capturing how well both global and local structures are *oriented* to each other, which is often falsely depicted and misleading in state-of-the-art approaches. The measure in (iii) reflects how well *local* structures are preserved, the typical objective in state-of-the-art approaches, whereas (iv) measures how well the *difference* (here, density) between local structures is preserved, penalizing for example when clusters of different size in $X$ are indistinguishable in size in $Y$.

For existing work which focuses on reconstructing local distances correctly, the neighborhood parameter is crucial for embedding quality. We investigated the impact of this parameter on each method considering above metrics and found that there is a clear trade-off between local and global reconstruction, with increasing neighborhood respectively perplexity showing better reconstruction of long-range relationships but much worse reconstruction of local features, also evident in the embeddings. Most notably, there was no hyperparameter setting that was consistently better across metrics – different neighborhood sizes are optimal for different metrics. We thus decided to stick to the recommended default if it was best for at least one metric, providing the analysis in App. C.5.

**Low-dimensional data** We first consider three datasets of 2 or 3 dimensions, as these can be directly visualized and hence compared to (see App. Sec. C.3 for details). They exemplify difficult issues of current methodology for low-dimensional embeddings. On simple data resembling a smiley face, a focus on reconstructing local distances results in the relative orientation of structures—here the eye, mouth and face outline—not being faithfully reconstructed and manifolds being distorted (see Fig. 2 top). On a real 3D manifold representing the reconstruction of a mammoth (The Smithsonian Institute, 2020; Wang et al., 2021), we investigate how well complex manifolds are preserved, observing that current methods have issues deriving a meaningful embedding of the original data (see Fig. 2 mid row). We observe the often discussed forming of "arbitrary" clusters in UMAP, TSNE, and NCVIS at varying degrees of intensity. Studying different settings of the neighborhood parameter for existing work shows a trade-off between local and global feature reconstruction; with small neighborhood we see more clustering but also better capturing of local features, larger neighborhoods give better global reconstruction at the cost of fine-grained features (see App. C.5). MERCAT produces a faithful embedding of the mammoth that captures not only the main features but the pose of the animal. For a simple circle in 2D, we observe that for such symmetric data current methods tend to break these symmetries deforming the circle and breaking it into clusters (see Fig. 2 bottom), which is consistent with the literature (Kobak & Linderman, 2021a). An exception is HMDS, as this data can be well represented by placing it all at a constant radius within the space—imagining an upward facing cone of a hyperbolic space, placing it exactly horizontally in that cone. We report all quantitative results in App. Tab. 3, observing that local neighborhoods are well preserved for existing methods, yet neither (global) distances nor angles are properly modeled. Both for a complex manifold as well as the highly symmetric circle data, the common objectives focusing on preserving local distances fail to yield faithful embeddings. MERCAT, on the other hand, yields embeddings that are as locally accurate as concurrent work, but outperforms them regarding distance and angle preservation (see Fig. 2, App. Tab. 3,4).

**Cluster data** To evaluate on synthetic data that is standard in the literature, such as Gaussian mixtures, we consider five different datasets, varying number of clusters, distribution type, number of sample and density per cluster (see App. Sec. C). We sample each dataset three times and report

Table 1: *Real data results.* We report angle preservation ($\angle$), distance preservation ($\|.\|$), neighborhood preservation ($\therefore$), and density preservation ($\odot$) between computed low-dimensional embeddings and original data. All numbers are rounded to two decimal places, higher is better, and **best method in bold**, second best underlined, "$-$" indicates method did not terminate within 24h.

| Data | Metric | MERCAT (ours) | UMAP | TSNE | NCVIS | DENSMAP | HMDS |
|------|--------|-----------|------|------|-------|---------|------|
| Tab. Sap. blood | $\angle$ | **.26** | .17 | .07 | .07 | .13 | $-$ |
| | $\|.\|$ | **.25** | .09 | $-.07$ | $-.01$ | .09 | $-$ |
| | $\therefore$ | .07 | .02 | **.29** | .02 | .02 | $-$ |
| | $\odot$ | **.22** | .00 | .11 | $-.16$ | .21 | $-$ |
| Murine Pancreas $n = 50$ | $\angle$ | **.48** | .33 | .46 | .34 | .40 | $-$ |
| | $\|.\|$ | **.61** | .41 | .45 | .32 | .48 | $-$ |
| | $\therefore$ | .10 | .07 | **.34** | .06 | .10 | $-$ |
| | $\odot$ | **.56** | $-.22$ | .01 | .14 | .27 | $-$ |
| Hematop. Paul et al. $n = 50$ | $\angle$ | **.86** | .76 | .82 | .34 | .77 | $NA$ |
| | $\|.\|$ | **.92** | .75 | .81 | .44 | .82 | .90 |
| | $\therefore$ | .31 | .28 | **.35** | .30 | .28 | .22 |
| | $\odot$ | **.66** | .29 | .08 | .11 | .63 | .58 |
| MNIST even $n = 50$ | $\angle$ | **.53** | .35 | .34 | .33 | .35 | $-$ |
| | $\|.\|$ | **.61** | .35 | .36 | .34 | .36 | $-$ |
| | $\therefore$ | .04 | .11 | **.20** | .11 | .10 | $-$ |
| | $\odot$ | .09 | $-.06$ | .14 | .10 | **.45** | $-$ |
| Cell Cycle $n = 50$ | $\angle$ | **.44** | .20 | .24 | .28 | .21 | $NA$ |
| | $\|.\|$ | .51 | .21 | .20 | .27 | .21 | **.61** |
| | $\therefore$ | .18 | .07 | **.27** | .07 | .11 | .09 |
| | $\odot$ | .35 | .37 | **.43** | .16 | .17 | .39 |

mean and standard deviation across different metrics in App. Tab. 3, 4, giving visualizations for a fixed random seed in Supp. Fig. 10, 11. We observe that, consistent with the literature (Kobak & Linderman, 2021a), neither UMAP nor TSNE consistently outperform the other. As expected, DENSMAP, which explicitly optimizes for recovering local densities, outperforms all other methods on the investigated data in terms of density preservation. Also, the general trends comparing between datasets are similar for all methods; the uniform data is more challenging than the simple Gaussian data (Unif5 vs Gauss5), and more clusters are harder to reconstruct (Gauss10 vs Gauss5). On the challenging Gauss5-*S* and Gauss5-*D* data which have strongly varying densities between clusters, MERCAT shows to be more robust than both TSNE and UMAP. Interestingly, MDS-based approaches, which also consider global reconstruction, perform well compared to TSNE and UMAP when computed with modern algorithmic tricks, already providing a good alternative to those LDE methods dominating the recent literature. Still, MERCAT performs better and, as we will see in the real-world data experiments next, scales to larger datasets. In summary, across experiments we see that MERCAT not only usually outperforms competitors in terms of angle preservation—which it was optimized for—but also overall distance reconstruction and, perhaps surprisingly, preservation of density in most cases, commonly ranking first or second for these metrics.

**Real world data** We evaluate the methods on three single-cell gene expression datasets of different origin resembling the most typical application of LDEs, in particular samples of human blood from the Tabula Sapiens project (The Tabula Sapiens Consortium, 2022), bone marrow in mice (Paul et al., 2015), and from the Murine Pancreas (Byrnes et al., 2018). We provide details on processing of the data in App. C.4. LDEs should capture the structure of blood cell differentiation. We further consider the MNIST (Lecun et al., 1998), where we focus on even numbers, as state-of-the-art methods are presumably good at clustering and should hence be able to capture these well-separated classes better. Lastly, we consider a dataset of cells with estimated cell cycle stage (Schwabe et al., 2020), an LDE can hence reflect the cyclic dependency of cell states. We report results in Tab. 1 and provide all visualizations in App. Sec. C.8.

Consistent with our previous findings, we see that MERCAT performs best in terms of angle and overall distance preservation. TSNE is best in reconstructing local neighborhoods, with MERCAT

usually taking second place. As expected, DensMAP outperforms existing work in terms of reconstruction of (local) densities in most cases, with cell cycle data being an exception. Perhaps surprisingly, Mercat performs best in three out of five datasets regarding density reconstruction, despite not explicitly modeling this property. In terms of quantitative results, Mercat seems to strike a balance of reconstructing both local as well as global structures also on real-world data.

Looking more closely into the Tabula Sapiens data (cf App. C.8), UMAP and DensMAP struggle with a proper fine-grained reflection of the data, as immune vs non-immune cells are dominating the overall structure and little structure is visible within immune cell clusters. tSNE learns several clusters, but dependencies between cell types are hard to make out. NCVis is able to find a more global structure, as well as differentiating locally between particular cell types, the visible global dependency looks, however, overly complex, much like the induced arbitrary bends on the Circle toy example (cf. Fig. 2d). Mercat learns a clearly visible and interpretable local and global structure reflecting relationships of different blood cell types, which together with the quantitative results indicate a more faithful reconstruction of the high-dimensional data. On MNIST, we see the known exaggeration of clustering by existing methods, which gives a clearer separation of digits. Mercat shows a greater mixture of cluster boundaries. While this sacrifices a bit of local reconstruction, it seems to better represent global relationships (cf. Tab. 1). For this particular dataset, we observe a strong trade-off between local and global structure preservation. Murine Pancreas as well as the human bone marrow data on a first glance look similar across methods, with all being able to distinguish cell types, encoding global dependencies that reflect hematopoiesis. Yet, tSNE and NCVis seem to have issues getting the long-range dependencies right, and all existing methods often show formations of seemingly arbitrary clusters. Neither C-Isomap nor MDS-based approaches did converge on these datasets within reasonable time ($\leq$ 1day), only yielding embeddings for cell cycle and the smaller hematopoiesis data of Paul et al. On cell cycle data, only Mercat and DensMAP are able to capture the cyclic structure of the data, correctly embedding the dependencies between the different cell cycle stages. All other methods are not reflecting the cell stage transitions properly. While the results of HMDS do look cyclic, we see that this is a side-effect of visualizing the hyperbolic space as points of all cell cycle stages are interspersed at every location of the circle.

## 5 Discussion and Conclusion

In this work, we suggested a new paradigm for the computation of low-dimensional embeddings, arguing for a simpler approach compared to current methodologies. The central question we ask is whether reconstructing primarily local features, as common in state-of-the-art, is what we want, given that this approach profoundly constrains the quality of reconstruction of the global properties of the data. Different from existing work, we cast the underlying optimization problem in terms of *reconstructing angles between any set of three points* correctly on a 2-dimensional sphere. We suggested an efficient approach called Mercat that can easily learn LDEs by off-the-shelf gradient descent optimizers. Further, we both empirically as well as theoretically motivate a sub-sampling approach and an initial denoising step, which improves the efficiency and robustness of the proposed algorithm for large and high-dimensional datasets. On synthetic, real-world, and easy-to-understand low-dimensional data, we show that our approach effectively recovers *both local as well as global structures*, outperforming existing methods despite, or maybe because of, its simplicity. It thus supports the hypothesis that the trade-offs between local and global reconstruction are caused by algorithm choice rather than theoretical limitation.

While giving highly encouraging results, our work also leaves room for future improvements. One direction of research could be further improvements of embedding quality; Mercat mostly outperforms existing work in terms of angle-, distance-, and neighborhood-preservation, yet is often seconded by DensMAP in terms of density preservation. While this may come by little surprise, as DensMAP explicitly optimizes for density preservation, it would still make for exciting future work to improve Mercat in that regard. Also, algorithmic advances targeting the efficiency could be interesting; the current methodology of Mercat is applicable to arbitrary sized datasets as it linearly scales with the number of samples thanks to the subsampling procedure, but is not ideal due to a large constant factor. For close to online performance on very large datasets, similar to NCVis (Artemenkov & Panov, 2020) or FasttSNE (Linderman et al., 2019), additional work is required. Lastly, we anticipate further theoretical insights, as the simple optimization loss lends itself for rigorous analysis.

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

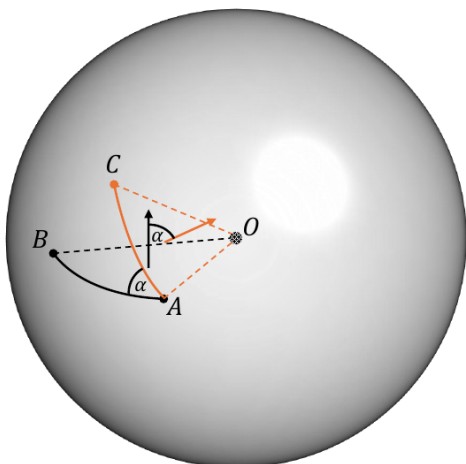

Figure 3: *Computing sphere angles with linear algebra.* We visualize the idea of computing the angle $\alpha$ between two (geodesic) paths $\overline{AB}, \overline{AC}$ on a sphere. The key insight is that the angle between the two geodesics is the same as the angle between the normals (visualized as arrows) of the two triangles $\Delta OAB, \textcolor{orange}{\Delta OAC}$ in the ambient 3D space, with $O$ as center of the sphere.

# A  ALGORITHM

## A.1  GEOMETRY ON SPHERE

*Side-to-angle formula (spherical law of cosines).* Let $\Delta ABC$ be a triangle on the sphere, with $a = \overline{BC}$, $b = \overline{AC}$, $c = \overline{AB}$, $\alpha = \angle(CAB)$, $\beta = \angle(CBA)$, and $\gamma = \angle(ACB)$. Then we have $\cos \alpha = \frac{\cos a - \cos b \cos c}{\sin b \sin c}$, $\cos \beta = \frac{\cos b - \cos c \cos a}{\sin c \sin a}$, $\cos \gamma = \frac{\cos c - \cos a \cos b}{\sin a \sin b}$.

*Vertex-to-side formula.* For $Y_i = (\phi_i, \theta_i)$ and $Y_j = (\phi_j, \theta_j)$ on the sphere, it follows that the geodesic distance between $Y_i$ and $Y_j$ is $\overline{Y_i Y_j} = d(Y_i, Y_j) = \sqrt{2 - 2[\sin \phi_i \sin \phi_j \cos(\theta_i - \theta_j) + \cos \phi_i \cos \phi_j]}$.

## A.2  COMPUTING GEODESIC ANGLES WITH LINEAR ALGEBRA

To efficiently compute geodesics on a sphere that is numerically stable and suitable for computation on graphics cards, we use the following trick.

To compute an angle $\angle BAC$ between the (geodesic) paths $\overline{AB}, \overline{AC}$ at point $A$, respecting the curvature of the sphere, we use the fact that the angle between these geodesics is the angle between the two planes $p_{OAB}$ and $p_{OAC}$ in the ambient 3D space, where $p_{ijk}$ is the plane that is spanned by the three points $i, j, k$, and $O$ is the center of the sphere, which we assume to be the origin of the space w.l.o.g.. Using the further insight that the angle between these two planes is the angle between their normal vectors, we can use the cross product to compute the two normal vectors, normalize the vectors to unit length and then compute the enclosed angle by using the definition of the scalar product

$$\angle BAC = \cos^{-1}\left( \frac{A \otimes B}{||A \otimes B||} \cdot \frac{A \otimes C}{||A \otimes C||} \right),$$

with $\otimes$ as the cross product. We provide a visualization of this idea in Fig. 3. In practice, as discussed in the main paper, we will drop the inverse cosine function in both high- and low-dimensional angle computations, which is a strictly monotone transformation.

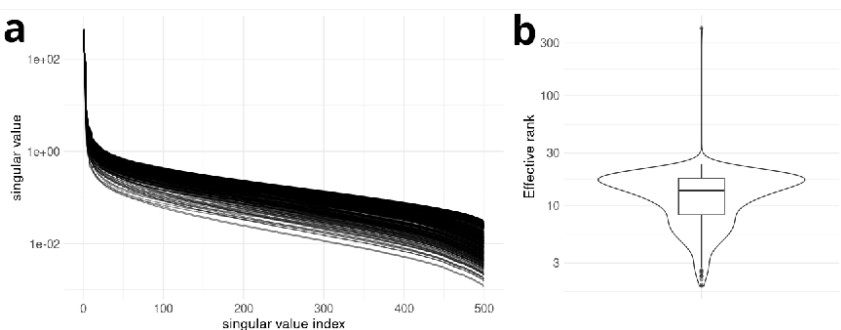

Figure 4: *Spectral analysis of angle space.* For 500 samples randomly taken from human hematopoiesis data Paul et al. (2015) we show (**a**) the singular values of the matrix $\Theta_i$ of cosine-angles at sample $i$ (one line per sample) and (**b**) the distribution of effective rank of all $\Theta_i$ on this dataset. Angle matrices are of low (effective) rank, thus encourage subsampling of angles.

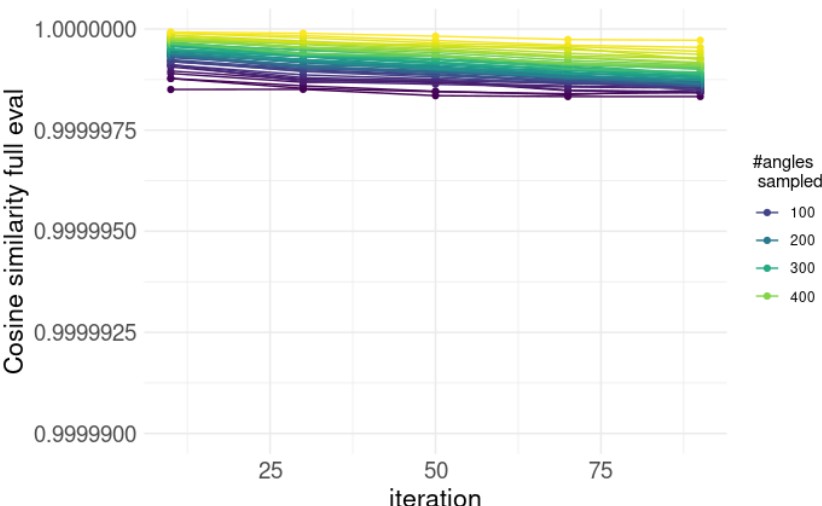

Figure 5: *Agreement between approximated and true gradient direction.* For 500 samples randomly taken from human hematopoiesis data Paul et al. (2015) we show the cosine similarity between the gradients obtained through subsampling of and the evaluation of all angles during the progress of optimization (x-axis). Colors indicate different number of subsampled angles. Agreement between subsampled and true gradients is virtually 1.

# B THEORY

## B.1 PROOF OF THEOREM 1

We define

$$f(\sigma_i) = \frac{(1 + \phi^{1/2}\sigma_i)(1 - \sigma_i^{-2})}{\phi^{1/2}\tau(\sigma_i)} \tag{5}$$

where $\tau(x) = \phi^{1/2} + \phi^{-1/2} + x + x^{-1}$. By definition, we have

$$\widehat{U}_i^\top \widehat{U}_j = \mathbf{e}_i^\top \widehat{U}\widehat{U}^\top \mathbf{e}_j, \qquad U_i^\top U_j = \mathbf{e}_i^\top U U^\top \mathbf{e}_j. \tag{6}$$

In the following lemma, proved in (Bloemendal et al., 2016, Section 5) and (Bao et al., 2022, Section 5), concerns the limiting behavior of the bilinear form $\mathbf{w}_1^\top \widehat{U}\widehat{U}^\top \mathbf{w}_2$ for any unit vectors $\mathbf{w}_1, \mathbf{w}_2 \in \mathbb{R}^n$.

**Lemma 1.** *Under the assumption of Theorem 1, for any unit vectors $\mathbf{w}_1, \mathbf{w}_2 \in \mathbb{R}^n$, it holds that*

$$\mathbf{w}_1^\top \widehat{U}\widehat{U}^\top \mathbf{w}_2 = \sum_{k=1}^r f(\sigma_k)\mathbf{w}_1^\top \mathbf{u}_k \mathbf{u}_k^\top \mathbf{w}_2 + O_P(n^{-1/2+\epsilon}), \tag{7}$$

*for any small constant $\epsilon > 0$, where $f(\sigma_k)$ is defined in (5).*

As a result, if we denote

$$u_{ki} = \mathbf{e}_i^\top \mathbf{u}_k, \qquad 1 \le i \le n, \quad 1 \le k \le r,$$

it then follows that

$$\mathbf{e}_i^\top \widehat{U}\widehat{U}^\top \mathbf{e}_j = \sum_{k=1}^r u_{ki} u_{kj} f(\sigma_k) + O_P(n^{-1/2+\epsilon}) = \mathbf{e}_i^\top U\Gamma U^\top \mathbf{e}_j + O_P(n^{-1/2+\epsilon})$$

where $\Gamma = \mathrm{diag}\,(f(\sigma_1), ..., f(\sigma_r))$. As a result, it follows that

$$|\mathbf{e}_i^\top \widehat{U}\widehat{U}^\top \mathbf{e}_j - \mathbf{e}_i^\top U U^\top \mathbf{e}_j| = |\mathbf{e}_i^\top U(\Gamma - I_r)U^\top \mathbf{e}_j| + O_P(n^{-1/2+\epsilon})$$

$$= \left| \sum_{k=1}^r u_{ki} u_{kj} \left( \frac{(1 + \phi^{1/2}\sigma_k)(1 - \sigma_k^{-2})}{\phi^{1/2}\tau(\sigma_k)} - 1 \right) \right| + O_P(n^{-1/2+\epsilon})$$

$$= \sum_{k=1}^r \frac{(1 + \phi^{1/2}\sigma_k)u_{ki} u_{kj}}{\sigma_k(\sigma_k + \phi^{1/2})} + O_P(n^{-1/2+\epsilon})$$

This completes the proof.

**Proof of Equation (4)**   Note that by Equation (2), as long as $\|U_j\|^2$ and $\|U_i\|^2$ are bounded away from 0, which is the case in our setup, $\theta_{jk,i}$ is a Lipschitz continuous function with respect to each of them. As a result, Equation (4) holds as long as the same statement hold for each of these terms. In other words, it suffices to show that for any $\epsilon' > 0$, there exist sufficiently large $(\sigma_1, ..., \sigma_r)$ such that

$$\lim_{n \to \infty} P(|\widehat{U}_i^\top \widehat{U}_j - U_i^\top U_j| \ge \epsilon') = 0.$$

This equation follows from Theorem 1, since $\sum_{k=1}^r \frac{(1+\phi^{1/2}\sigma_k u_{ki} u_{kj}}{\sigma_k(\sigma_k + \phi^{1/2})}$ is monotonic decreasing function of $\sigma_k$, which can be made arbitrarily close to 0 when $(\sigma_1, ..., \sigma_r)$ increase. This completes the proof.

## B.2 PROOF OF THEOREM 2

Note that $\Sigma = I + \phi \sum_{s=1}^r \sigma_s \mathbf{u}_s \mathbf{u}_s^\top$ implies

$$\Sigma_{ij} = \phi \sum_{s=1}^r \sigma_s u_{si} u_{sj} + \delta_{ij}.$$

Then we have

$$\frac{\Sigma_{jk} - \Sigma_{ik} - \Sigma_{ji} + \Sigma_{ii}}{\sqrt{\Sigma_{jj} + \Sigma_{ii} - 2\Sigma_{ji}}\sqrt{\Sigma_{kk} + \Sigma_{ii} - 2\Sigma_{ki}}} = \frac{\phi \sum_{s=1}^{r} \sigma_s(u_{sj}u_{sk} - u_{si}u_{sk} - u_{si}u_{sj} + u_{si}^2) + 1}{\phi\sqrt{\sum_{s=1}^{r} \sigma_s(u_i - u_j)^2}\sqrt{\sum_{s=1}^{r} \sigma_s(u_i - u_k)^2}}$$

$$= \frac{(U_j - U_i)^\top W(U_k - U_i) + (\phi\sigma_r)^{-1}}{\sqrt{(U_j - U_i)^\top W(U_j - U_i)}\sqrt{(U_k - U_i)^\top W(U_k - U_i)}},$$

where $W = \mathrm{diag}(\sigma_1/\sigma_r, \sigma_2/\sigma_r..., 1)$. If we denote

$$\boldsymbol{\beta} = U_j - U_i, \qquad \boldsymbol{\gamma} = U_k - U_i,$$

and

$$\widetilde{\boldsymbol{\beta}} = W^{1/2}(U_j - U_i), \qquad \widetilde{\boldsymbol{\gamma}} = W^{1/2}(U_k - U_i),$$

it follows that

$$\frac{\Sigma_{jk} - \Sigma_{ik} - \Sigma_{ji} + \Sigma_{ii}}{\sqrt{\Sigma_{jj} + \Sigma_{ii} - 2\Sigma_{ji}}\sqrt{\Sigma_{kk} + \Sigma_{ii} - 2\Sigma_{ki}}} = \frac{\widetilde{\boldsymbol{\beta}}^\top\widetilde{\boldsymbol{\gamma}} + (\sigma_r\phi)^{-1}}{\|\widetilde{\boldsymbol{\beta}}\|\|\widetilde{\boldsymbol{\gamma}}\|} \tag{8}$$

On the other hand, we have

$$\cos\theta_{jk,i} = \frac{(U_j - U_i)^\top(U_k - U_i)}{\|U_j - U_i\|\|U_k - U_i\|}$$

$$= \frac{\boldsymbol{\beta}^\top\boldsymbol{\gamma}}{\|\boldsymbol{\beta}\|\|\boldsymbol{\gamma}\|}.$$

Now if we denote $\theta = \angle(\boldsymbol{\beta}, \boldsymbol{\gamma})$ and $\widetilde{\theta} = \angle(\widetilde{\boldsymbol{\beta}}, \widetilde{\boldsymbol{\gamma}})$, it follows that

$$\left|\frac{\widetilde{\boldsymbol{\beta}}^\top\widetilde{\boldsymbol{\gamma}} + (\sigma_r\phi)^{-1}}{\|\widetilde{\boldsymbol{\beta}}\|\|\widetilde{\boldsymbol{\gamma}}\|} - \frac{\boldsymbol{\beta}^\top\boldsymbol{\gamma}}{\|\boldsymbol{\beta}\|\|\boldsymbol{\gamma}\|}\right| \geq |\cos\widetilde{\theta}| - |\cos\theta| - \frac{1}{\phi\sigma_r\|\boldsymbol{\beta}\|\|\boldsymbol{\gamma}\|},$$

where in the last inequality we used $\|\boldsymbol{\beta}\| \leq \|\widetilde{\boldsymbol{\beta}}\|$ and $\|\boldsymbol{\gamma}\| \leq \|\widetilde{\boldsymbol{\gamma}}\|$. To obtain the final result, we first show that, there exists some $W$ so that $\cos\widetilde{\theta}$ can be made arbitrarily close to $1$ or $-1$. Without loss of generality, we assume $\cos\theta > 0$, and $\beta_1\gamma_1 \neq 0$, where we use the notation $\boldsymbol{\beta} = (\beta_1, ..., \beta_r)$ and $\boldsymbol{\gamma} = (\gamma_1, ..., \gamma_r)$. Moreover, we denote $\alpha = \frac{1 - \cos\theta}{\cos\theta}$ so that $1 = (1 + \alpha)\cos\theta$. Now if $\beta_1\gamma_1 > 0$, then we can always find $W$ so that $\sigma_1/\sigma_r$ is significantly larger than $\{\sigma_2/\sigma_r, ..., 1\}$, and therefore either

$$\max\{\angle(\widetilde{\boldsymbol{\beta}}, \mathbf{e}_1), \angle(\widetilde{\boldsymbol{\gamma}}, \mathbf{e}_1)\} < \frac{1}{2}\arccos\left(\left(1 + \frac{\alpha}{2}\right)\cos\theta\right)$$

or

$$\max\{\angle(\widetilde{\boldsymbol{\beta}}, -\mathbf{e}_1), \angle(\widetilde{\boldsymbol{\gamma}}, -\mathbf{e}_1)\} < \frac{1}{2}\arccos\left(\left(1 + \frac{\alpha}{2}\right)\cos\theta\right)$$

holds. In either case, we have

$$\angle(\widetilde{\boldsymbol{\beta}}, \widetilde{\boldsymbol{\gamma}}) < \arccos\left(\left(1 + \frac{\alpha}{2}\right)\cos\theta\right)$$

so that

$$\cos\widetilde{\theta} > \left(1 + \frac{\alpha}{2}\right)\cos\theta.$$

If instead $\beta_1\gamma_1 < 0$, then we can similarly choose $\sigma_1/\sigma_r$ sufficiently larger than $\{\sigma_2/\sigma_r, ..., 1\}$ so that either

$$\max\{\angle(\widetilde{\boldsymbol{\beta}}, \mathbf{e}_1), \angle(\widetilde{\boldsymbol{\gamma}}, -\mathbf{e}_1)\} < \frac{1}{2}\arccos\left(\left(1 + \frac{\alpha}{2}\right)\cos\theta\right)$$

or

$$\max\{\angle(\widetilde{\boldsymbol{\beta}}, -\mathbf{e}_1), \angle(\widetilde{\boldsymbol{\gamma}}, \mathbf{e}_1)\} < \frac{1}{2}\arccos\left(\left(1 + \frac{\alpha}{2}\right)\cos\theta\right)$$

holds. In either case, we have

$$\angle(\widetilde{\boldsymbol{\beta}}, \widetilde{\boldsymbol{\gamma}}) > \pi - \arccos\left(\left(1 + \frac{\alpha}{2}\right)\cos\theta\right),$$

so that

$$\cos \widetilde{\theta} < -\left(1 + \frac{\alpha}{2}\right) \cos \theta.$$

As a result, we have

$$\left| \frac{\widetilde{\boldsymbol{\beta}}^\top \widetilde{\boldsymbol{\gamma}} + (\sigma_r \phi)^{-1}}{\|\widetilde{\boldsymbol{\beta}}\|\|\widetilde{\boldsymbol{\gamma}}\|} - \frac{\boldsymbol{\beta}^\top \boldsymbol{\gamma}}{\|\boldsymbol{\beta}\|\|\boldsymbol{\gamma}\|} \right| \geq \frac{\alpha}{2} \cos \theta - \frac{1}{\phi \sigma_r \|\boldsymbol{\beta}\|\|\boldsymbol{\gamma}\|}. \tag{9}$$

Finally, with $W$ and $U$ fixed and $\phi$ bounded away from zero, we can always choose sufficiently large $\sigma_r > 0$ such that

$$\frac{1}{\phi \sigma_r \|\boldsymbol{\beta}\|\|\boldsymbol{\gamma}\|} < \frac{\alpha}{4} \cos \theta.$$

Combining the above results, we have

$$\left| \frac{\Sigma_{jk} - \Sigma_{ik} - \Sigma_{ji} + \Sigma_{ii}}{\sqrt{\Sigma_{jj} + \Sigma_{ii} - 2\Sigma_{ji}}\sqrt{\Sigma_{kk} + \Sigma_{ii} - 2\Sigma_{ki}}} - \cos \theta_{jk,i} \right| \geq \frac{\alpha}{4} \cos \theta, \tag{10}$$

or

$$\left| \arccos \left( \frac{\Sigma_{jk} - \Sigma_{ik} - \Sigma_{ji} + \Sigma_{ii}}{\sqrt{\Sigma_{jj} + \Sigma_{ii} - 2\Sigma_{ji}}\sqrt{\Sigma_{kk} + \Sigma_{ii} - 2\Sigma_{ki}}} \right) - \theta \right| \geq \delta, \tag{11}$$

for some constant $\delta > 0$ only depending on $\theta$. Finally, it suffices to note that, by the law of large numbers and the continuous mapping theorem, we have

$$\arccos \left( \frac{(X_j - X_i) \cdot (X_k - X_i)}{\|X_j - X_i\|\|X_k - X_i\|} \right) \to_P \arccos \left( \frac{\Sigma_{jk} - \Sigma_{ik} - \Sigma_{ji} + \Sigma_{ii}}{\sqrt{\Sigma_{jj} + \Sigma_{ii} - 2\Sigma_{ji}}\sqrt{\Sigma_{kk} + \Sigma_{ii} - 2\Sigma_{ki}}} \right). \tag{12}$$

This along with (11) completes the proof of the theorem.

### B.3 EFFICACY OF SUBSAMPLING

Here, we provide some theoretical insights that partially explains the efficacy of our subsampling procedure. Recall that at each optimization iteration, for each data point $i$, instead of using of all the entries in the angle matrix $M_i = (\angle jik)_{1 \leq j,k \leq n}$, we only take a random subset of the entries. Our hope is that such a random subset contains sufficient information about the whole matrix. This is in the same spirit as the matrix completion problem where the goal is to recover the missing matrix entries from a small number of randomly observed entries (Candes & Plan, 2010; Keshavan et al., 2010; Cai et al., 2010; Candes & Recht, 2012). From theory of matrix completion, a critical condition enabling precise local-to-global reconstruction is known as the incoherence condition, which essentially requires that the matrix is approximately low-rank and its leading singular vectors are relatively "spread out," effectively avoiding any outliers in the data matrix. In our case, the spiked population model automatically implies the approximate low-rankness of the cosine-angle matrix $\widehat{\Theta}_i = (\widehat{\theta}_{jk,i})_{1 \leq j \neq k \leq n}$, which follows from (2) and that

$$\Theta_i \equiv (\theta_{jk,i})_{1 \leq j,k \leq n} = \left( \frac{(U_j - U_i)^\top (U_k - U_i)}{\|U_j - U_i\|\|U_k - U_i\|} \right) = D^{-1/2} V V^\top D^{-1/2}, \tag{13}$$

where

$$V = \begin{bmatrix} (U_1 - U_i)^\top \\ (U_2 - U_i)^\top \\ \dots \\ (U_n - U_i)^\top \end{bmatrix} \in \mathbb{R}^{n \times r}, \quad D = \mathrm{diag}(\|U_1 - U_i\|^2, ..., \|U_n - U_i\|^2), \tag{14}$$

showing that $\Theta_i$ has rank at most $r$. If we denote $W \in \mathbb{R}^{n \times r}$ as the matrix of singular vectors of $\Theta_i$, the incoherence condition amounts to saying that

$$\left\| W W^\top - \frac{r}{n} I_n \right\|_{\max} \leq \mu \frac{\sqrt{r}}{n} \tag{15}$$

for some small constant $\mu > 0$, where $\|(a_{ij})\|_{\max} = \max_{i,j} |a_{ij}|$. In particular, the incoherence condition (15) is likely satisfied if the low-dimensional signal structure with respect to the $i$th data point, encoded by $\{U_j - U_i\}_{1 \leq j \leq n}$, has certain smoothness property and does not contain outliers deviating significantly from the bulk, which is the case for many applications. For example, in typical biological applications an outlier removal is part of the preprocessing pipeline (Luecken & Theis, 2019; Heumos et al., 2023).

## C  EXPERIMENTS

### C.1  UMAP EMBEDDINGS ON THE SPHERE

To embed data points on the 2D sphere instead of 2D plane, we use the haversine distance metric with the standard UMAP algorithm, which yields an embedding specifying longitude and latitude on the 2D sphere. We closely followed the suggestion in the original UMAP implementation https://umap-learn.readthedocs.io/en/latest/embedding_space.html#spherical-embeddings.

### C.2  COMPUTATION OF EVALUATION METRICS

In the following, we provide an overview on how the evaluation metrics are defined.

- *distance preservation* ($||.||$) measured as Spearman Rank correlation coefficient between high- and low-dimensional distances, capturing how well overall structure is preserved. Distances for MERCAT embeddings are computed from geodesics on the sphere.

- *neighborhood preservation* ($\therefore$) as measured by the mean jaccard index of the $k$-nearest neighbors (here, $k = 50$) in high- and low-dimensional space across all points, $1/n \sum_i \frac{|knn(X,i) \cap knn(Y,i)|}{|knn(X,i) \cup knn(Y,i)|}$, where $knn(X,i)$ gives the indices of the $k$ nearest neighbors in $X$, capturing how accurate local structures are embedded. Before neighborhood computation, we denoise using ScreeNOT Donoho et al. (2023).

- *density preservation* ($\odot$), which reflects how well differences in densities are captured in the embedding, a recent point of interest in the literature Narayan et al. (2021); Fischer et al. (2023). We measure this by comparing the number of points that fall in spheres of constant radius around each point. More concretely, we compute the average distance of the $25th$-nearest neighbor in high- and low-dimensional space, $\bar{k}_{high}$ and $\bar{k}_{low}$, and for each sample $i$ compute the local density as number of points that fall into a sphere centered at $i$ of radius $\bar{k}_{high}$ resp. $\bar{k}_{low}$. The Pearson correlation coefficient between the obtained sphere densities gives our final metric.

- *preservation of angles* ($\angle$) between any three points measured as the Pearson Correlation coefficient between angles in high- and low-dimensional space, which captures how well global relationships, such as orientation of clusters are preserved. For practical purposes, as this computation is cubic in the number of points, we again sample for each point $i$ 64 other points at random and compute the angle at $i$ and all combination of other points.

We further investigated the effect of the neighborhood parameter for neighborhood and density scores across all methods, noting that relative order of methods is quite stable, except for TSNE and UMAP which perform well in the smaller neighborhood regime (including our chosen parameters) but much worse for larger neighborhood sizes. This comes to little surprise, as these methods are good at reconstructing local structure (small $k$) but bad in reconstructing global structure (large $k$). MERCAT performs well across different scales of $k$. The exemplary analysis on the cell cycle dataset with varying $k$ can be found in Fig. 6.

### C.3  REPRODUCIBILITY – GENERATION OF DATA

SMILEY

To obtain the Smiley dataset, we sample $n = 3000$ points as follows. A quarter of these points are used for the eyes, where we first draw a radius for each point as $e'_r \sim U(0,1)$ and further transform this radius to get $e_r = .1\sqrt{e'_r}$. We additionally draw an angle $e_\theta \sim U(0, 2\pi)$. The actual points are then assigned to the 2D coordinates $x = e_r \sin(e_\theta)$, $y = e_r * \cos(e_\theta)$. Half of these samples are then offset by $(.25, .25)$, the other half by $(-.25, .25)$, resulting in the final coordinates of the eyes. For the face outline we dedicate half of the overall points, first sampling a radius $f'_r \sim U(.9^2, 1)$, which is transformed to get $f_r = \sqrt{f'_r}$. We further draw an angle $f_\theta \sim U(0, 2\pi)$ and compute the final coordinates as $x = f_r \sin(f_\theta)$, $y = f_r * \cos(f_\theta)$. Lastly, we dedicate the remaining (quarter of) points to the mouth, sampling $m'_r \sim U(.45^2, .55^2)$, which is transformed

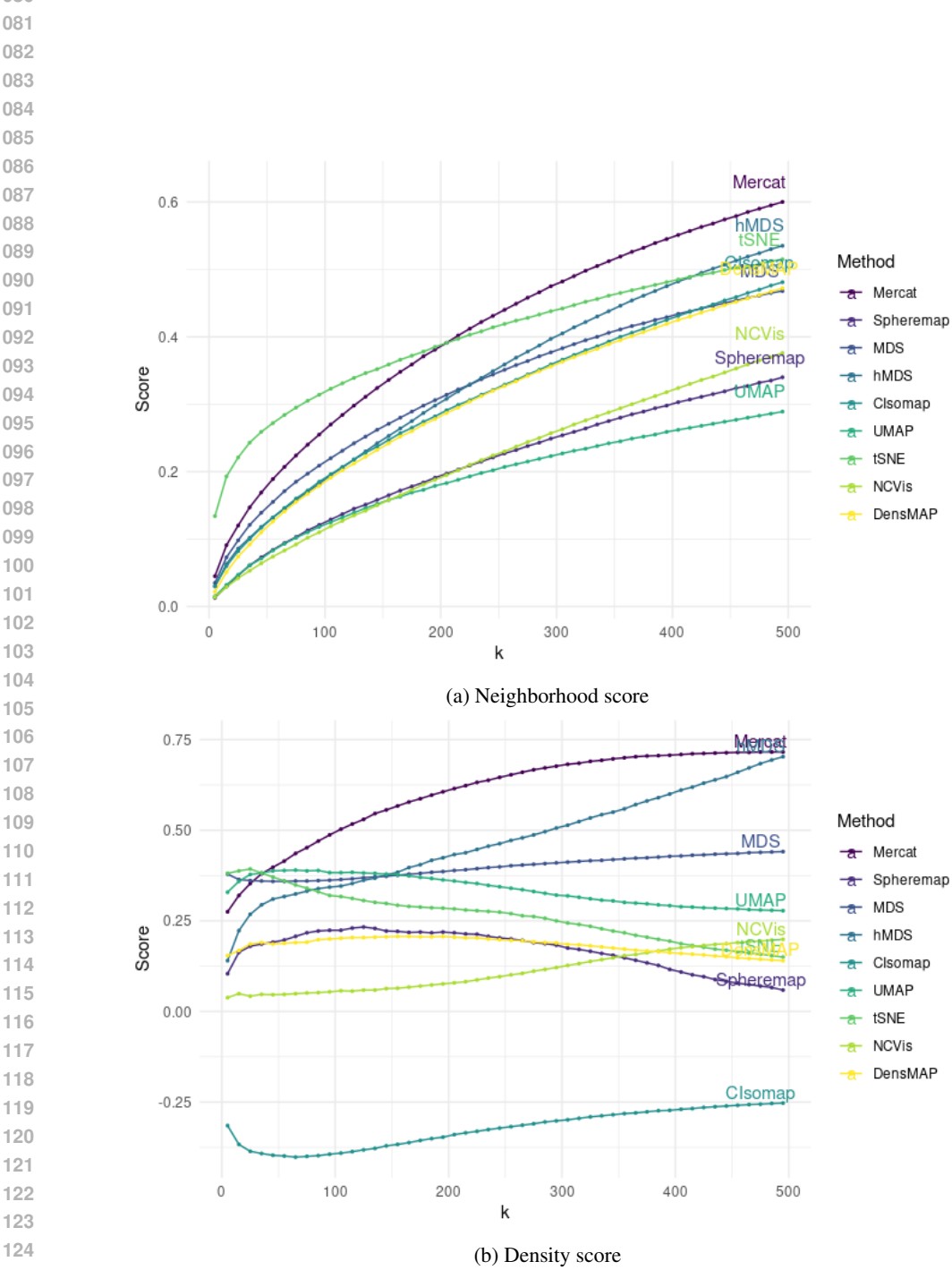

(a) Neighborhood score

(b) Density score

Figure 6: *Scores across varying neighborhood sizes.* We show the obtained scores of different methods on cell cycle data in terms of neighborhood (top) and density (bottom) reconstruction considering different neighborhood sizes $k$ for the score computation.

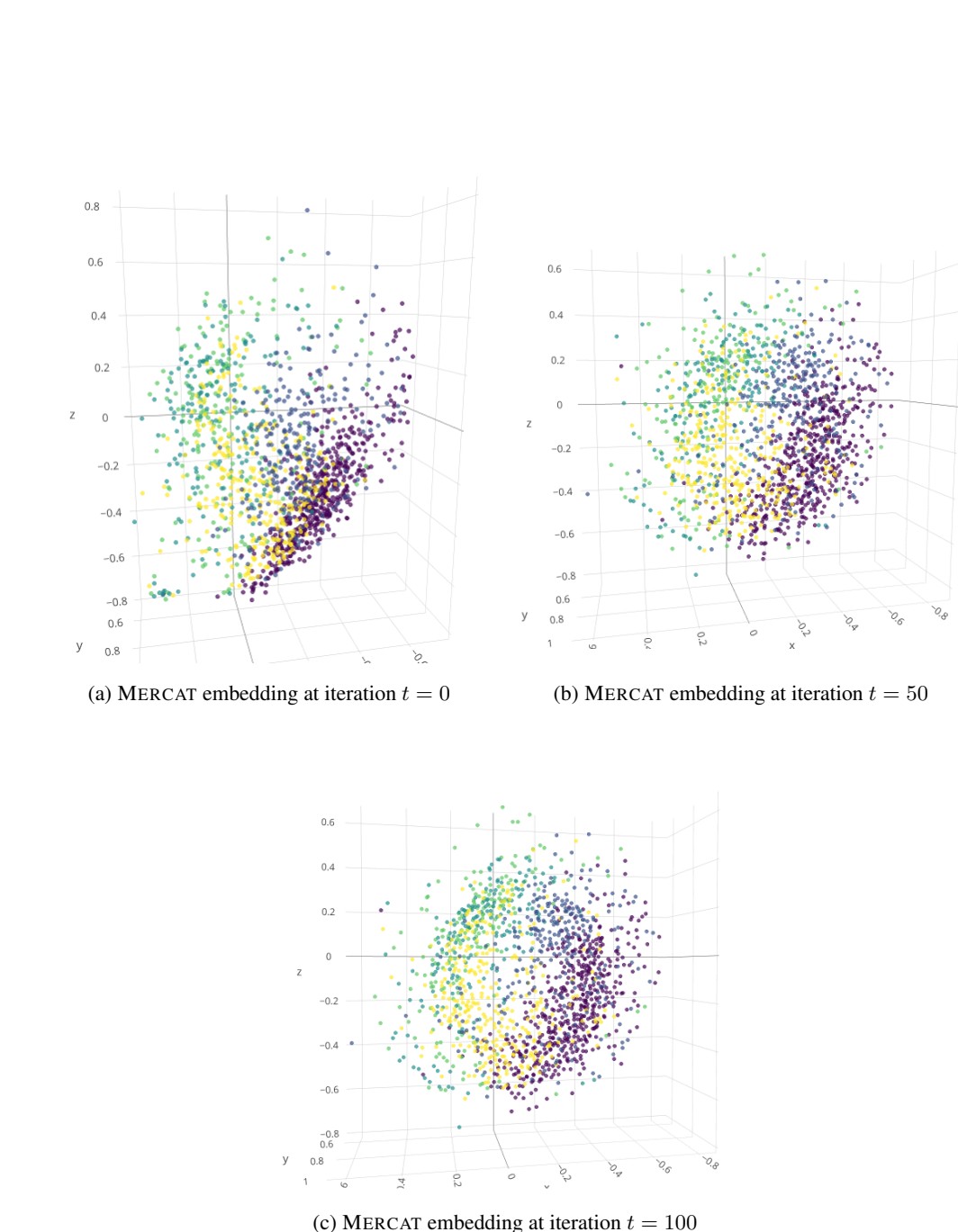

(a) MERCAT embedding at iteration $t = 0$

(b) MERCAT embedding at iteration $t = 50$

(c) MERCAT embedding at iteration $t = 100$

Figure 7: *Arrangement of points over optimization time.* We show the embedding of points through-out optimization progress for the cell cycle data, starting with the PCA embedding at initialization $t = 0$.

to get $m_r = \sqrt{m'_r}$. We further draw an angle $m_\theta \sim U(0, \pi)$ and compute the 2D coordinates as $x = m_r \sin(m_\theta),\ y = -m_r * \cos(m_\theta)$. Lastly, we scale the whole data by 2, concluding the data generation process

### CIRCLE

For the Circle data, we sample $n = 900$ angle $c_\theta \sim U(0, 2\pi)$ and compute the original circle as $x = 3\cos(c_\theta),\ y = 3 * \sin(c_\theta)$. We then add iid noise sampled from $N(0, .01)$ to both dimensions.

### GENERATION OF SYNTHETIC DATA

We generate (i) Unif5, a dataset in 50 dimensions of 5 uniform clusters with 100 samples each, with each dimension iid from $U(0, 1)$ and different centers sampled from $U(-10, 10)$, (ii) Gauss5, a dataset in 50 dimensions from 5 Gaussians with mean $\mu$ sampled from $U(-10, 10)$ (iid for each dimension) and standard deviation $\sigma$ sampled from $U(.5, 2)$ (iid for each dimension, all dimensions have covariance of 0), each cluster having 100 samples each, (iii) Gauss10, a dataset in 50 dimensions from 10 Gaussians with mean $\mu$ sampled from $U(-10, 10)$ (iid for each dimension) and standard deviation $\sigma$ sampled from $U(.5, 2)$ (iid for each dimension, all dimensions have covariance of 0), each cluster having 100 samples each, (iv) Gauss5-*S*, which is generated similar as Gauss5, but with different number of samples per cluster, namely 50,100,150,200, and 250 samples, and (v) Gauss5-*D*, which is generated similar as Gauss5, but with different densities per cluster using a covariance matrix as a diagonal matrix where entries are set 1, 2, 3, 4, and 5 for each cluster respectively.

### C.4 REPRODUCIBILITY – PREPROCESSING OF REAL DATA

**Tabula Sapiens human blood**   We obtained the human blood samples from the Tabula Sapiens project through the CZ CELLxGENE portal, preprocessed as Seurat object. We proceeded by filtering for data from the 10x 3' v3 assay to avoid strong batch effects due to different sequencing platforms. To filter for protein-coding genes – excluding genes encoded in the mitochondrium – we used the Gencode v38 genome annotation. We further filtered for genes that were expressed in at least one sample (i.e., sum of gene expression across samples was greater than zero). The annotated cell type in the data object was used for labeling.

**Murine pancreas**   We obtained pre-processed single-cell gene expression data through the Gene Expression Omnibus (accession id GSE132188). To filter for protein-coding genes, we used the genome annotation GRCm39.110. As before, we further filtered for genes that were expressed in at least one sample. For cell annotation, we use the provided clusters used in Figure 3 of the original publication Byrnes et al. (2018).

**Mouse bone marrow**   We obtained the pre-processed single-cell data of Paul et al. Paul et al. (2015) from the PAGA repository[2] Wolf et al. (2019).

**Cell cycle data**   The HeLa cell cycle annotated data was obtained following the github repository[3] of the original authors Schwabe et al. (2020), using the estimated phase as labels.

### C.5 HYPERPARAMETER CHOICES

We checked different hyper-parameter settings for existing work, focusing on varying the neighborhood respectively perplexity scores for UMAP, TSNE, NCVIS, and DENSMAP, as this is known to be one of the most deciding factors of embedding quality Kobak & Berens (2019). As datasets, we consider a representative subset using Unif5 from the cluster datasets, Mammoth from the low-dimensional manifold datasets, for both of which we vary the parameter $\theta \in 15, 30, 50, 100, 200$, and hematopoiesis data of Paul et al. from the real world datasets, for which we consider $\theta \in 15, 30, 100, 200, 500$, as it is considerably larger. We give the quantitative results in Tab. 2

---

[2] https://github.com/theislab/paga
[3] https://github.com/danielschw188/Revelio

and provide visualization of the mammoth reconstructions in Fig. 8, as we can compare these with the visualization of the original data (cf Fig. 2).

Across data, we see that quantitatively **there is no single best parameter** $\theta$, not across datasets, but more importantly, not within a dataset: varying the locality parameter $\theta$ (neighborhood or perplexity) means trading off local reconstruction performance against global reconstruction performance. This also becomes evident in the visualizations for mammoth (Fig. 8), where for UMAP and DENSMAP, which arguably give better reconstructions than competing methods, at smaller neighborhood size parameters the shape of the hip or leg bones as well as ribcage are still visible, at higher resolution the overall global structure looks like a more natural animal pose (albeit still wrong). We, hence, decided to use the **recommended default neighborhood parameter if at least one metric was "optimal"** during our evaluation. All other parameters were kept at their default value, noting that training converged in all but one case. This particular case was UMAP on the Tabula Sapiens blood data, where training with the default parameter yielded a particularly bad, artifacted visualization (albeit decent performance on local reconstruction). We then decided to set the neighborhood parameter to 50 to arrive at a meaningful embedding. For all remaining experiments we use the following setting:

UMAP **n_neighbor=15 (recommended default)**; use spectral initialization; $min\_dist = 0.1$;

TSNE **perplexity = 30 (recommended default)**; $initial\_dims = 50$; $theta = 0.5$; use PCA initialization; $max\_iter = 1000$; normalize data; $momentum = 0.5$; $final\_momentum = 0.8$; $eta = 200$; $exaggeration\_factor = 12$

NCVIS **n_neighbors=15 (recommended default)**; $n\_epochs = 50$; $n\_init\_epochs = 20$; $min\_dist = 0.4$

DENSMAP **n_neighbors = 30 (recommended default)**; spectral initialization, $dens\_frac = 0.3$; $dens_lambda = 0.1$; $dens\_var\_shift = 0.1$; $n\_epochs = 750$; $learning\_rate = 1$; $min\_dist = 0.1$

SPHEREMAP same as UMAP

NMMDS We use the euclidean distance for distance and dissimilarity computation. We use 20 random starts for the search for a stable solution. For more information, we refer to the `vegan` package.[4]

HMDS $curvature = 1$ (curvature of the space), equi-angular adjustment= .5 (adjusts data so that their angular coordinates are unif. distr. in the Poincare disc - otherwise circle and other datasets would be strongly distorted), $\alpha = 1.1$ (adjusts distortion of the embedding). All of these are recommended defaults.

C-ISOMAP Using centering of data for pre-processing and .1 of sample size as neighborhood size for graph construction (default).

For MERCAT, we use the standard parameters for the Adam optimizer as recommended in the original paper Kingma & Ba (2015). Throughout all experiments we set the initial learning rate to 0.01, and have a multiplicative learning rate schedule $\gamma$, multiplying by 0.1 at predefined iterations (i.e., reducing the learning rate by an order of magnitude). As discussed in the main paper, we use an angle subsampling of 64, and a batch size of 64. For all synthetic and toy experiments, we run for $t = 1000$ iterations, with a learning rate change at $\gamma = [350]$.

For real world data we reduce the number of iterations, as we do a batched learning approach and hence need much fewer iterations to see the same number of samples (and hence angles) as in the synthetic case studies. In particular, for MNIST we use $t = 250, \gamma = [100]$, for Tabula Sapiens and Murine Pancreas we use $t = 50, \gamma = [10, 30]$, for human bone marrow and cell cycle data we use $t = 200, \gamma = [50, 150]$.

*Note that in principle it is possible to optimize hyperparameters such as batch size, subsampling, etc to further improve MERCAT embeddings by calibrating based on angle reconstruction. We instead wanted to keep parameters constant across experiments to show MERCAT's wide applicability with a standard set of parameters and only vary the number of iterations and learning rate schedule linked to these iterations.*

---

[4]https://cran.r-project.org/web/packages/vegan/index.html

Table 2: *Results on different neighborhood parametrization.* We report angle preservation (∠), distance preservation (‖·‖), neighborhood preservation (∴), and density preservation (⊙) between computed low-dimensional embeddings and original data. All numbers are rounded to two decimal places, higher is better, and **best result is in bold.**

| Data | Metric | UMAP 15 | 30 | 50 | 100 | 200 | TSNE 15 | 30 | 50 | 100 | 200 | NCVIS 15 | 30 | 50 | 100 | 200 | DENSMAP 15 | 30 | 50 | 100 | 200 |
|---|---|---|---|---|---|---|---|---|---|---|---|---|---|---|---|---|---|---|---|---|---|
| Unit5 | ∠ | .45 | .42 | .45 | .44 | **.46** | .49 | .49 | .47 | .49 | **.57** | .52 | .54 | .52 | **.55** | .54 | **.48** | **.48** | .46 | **.49** | .48 |
| | ‖·‖ | .26 | .23 | .18 | .36 | **.38** | .35 | **.52** | .36 | .28 | .29 | .50 | .64 | .60 | .68 | **.83** | .43 | .19 | .20 | .39 | **.48** |
| | ∴ | **.16** | **.16** | **.16** | **.16** | **.16** | .16 | .16 | .17 | .17 | **.27** | **.17** | **.17** | **.17** | .16 | .14 | **.17** | .16 | **.17** | **.17** | .16 |
| | ⊙ | **.13** | .09 | .07 | .04 | .00 | .37 | .49 | .64 | **.75** | .18 | .42 | .46 | .48 | **.58** | -.69 | .44 | .51 | .56 | .60 | **.61** |
| Mammoth | ∠ | .59 | .62 | .61 | **.70** | **.70** | .28 | .50 | .57 | .69 | **.73** | .15 | .26 | .30 | .62 | **.75** | .63 | .68 | .69 | **.71** | **.71** |
| | ‖·‖ | .77 | .82 | .82 | .86 | **.87** | .38 | .61 | .65 | .77 | **.81** | .21 | .40 | .44 | .60 | **.79** | .83 | .86 | .88 | .90 | **.91** |
| | ∴ | .58 | **.59** | **.59** | .58 | .55 | .59 | .65 | .67 | **.69** | .66 | .54 | **.63** | .62 | .54 | .45 | .59 | **.60** | .59 | .57 | .56 |
| | ⊙ | **.06** | .01 | .03 | .02 | .05 | .09 | **.10** | .05 | .01 | .02 | **.35** | .20 | .07 | -.05 | -.04 | .71 | **.75** | .71 | .66 | .58 |
| Hematop. | ∠ | .75 | .745 | **.76** | **.76** | .75 | .73 | **.81** | .80 | .78 | .77 | .51 | .55 | .74 | .75 | **.77** | .76 | .74 | .75 | **.77** | **.77** |
| | ‖·‖ | .73 | **.74** | **.74** | **.74** | .73 | .73 | **.80** | **.80** | .78 | .77 | .55 | .58 | **.71** | .68 | .69 | .81 | .81 | **.83** | .82 | **.83** |
| | ∴ | **.28** | **.28** | **.28** | **.28** | **.28** | .33 | .35 | **.37** | **.37** | .35 | .30 | **.31** | .30 | .27 | .16 | .28 | .28 | .28 | **.29** | .28 |
| | ⊙ | **.31** | .27 | .23 | .23 | .23 | .12 | .06 | .08 | .10 | **.23** | **.21** | .14 | -.39 | -.44 | -.43 | .60 | .64 | .67 | .65 | **.68** |

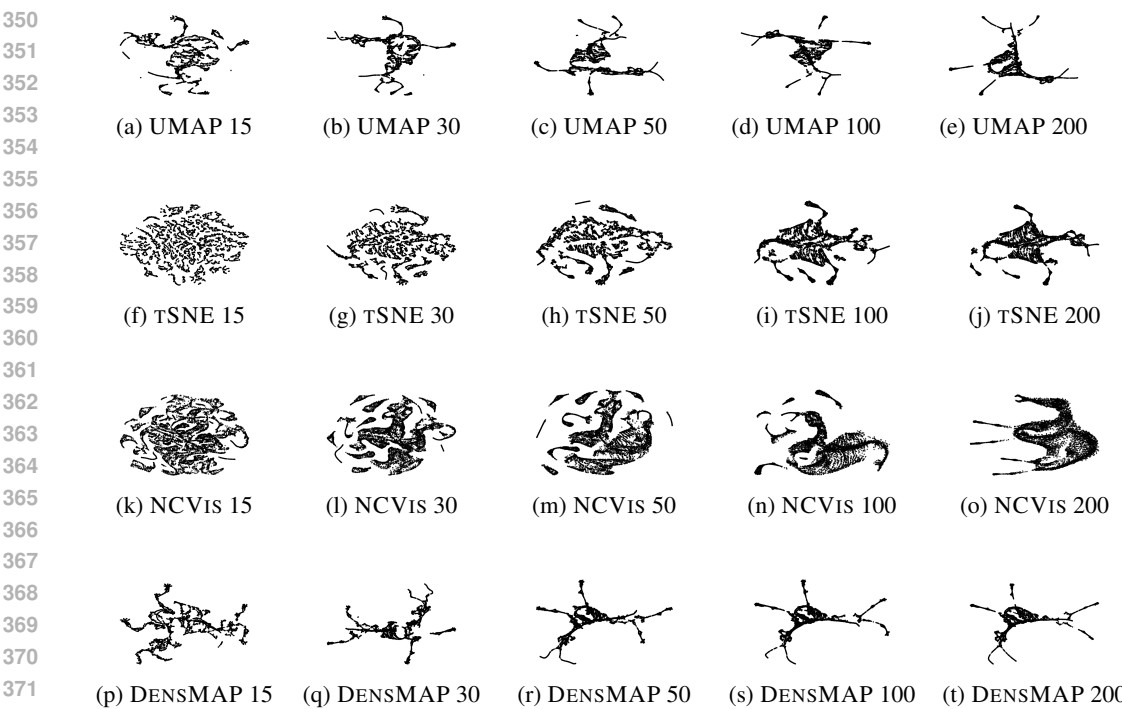

(a) UMAP 15    (b) UMAP 30    (c) UMAP 50    (d) UMAP 100    (e) UMAP 200

(f) TSNE 15    (g) TSNE 30    (h) TSNE 50    (i) TSNE 100    (j) TSNE 200

(k) NCVIS 15    (l) NCVIS 30    (m) NCVIS 50    (n) NCVIS 100    (o) NCVIS 200

(p) DENSMAP 15    (q) DENSMAP 30    (r) DENSMAP 50    (s) DENSMAP 100    (t) DENSMAP 200

Figure 8: *Embeddings for Mammoth with varying neighborhood size.* Visualizations for the Mammoth datasets for various neighborhood parameter setting for existing work, using neighborhood/perplexity scores of $\theta \in \{10, 20, 50, 100, 200\}$.

## C.6 VISUALIZATION-OPTIMAL ROTATIONS FOR 2D CONFORMAL MAPS

For a MERCAT embedding, for any rotation or translation on the sphere, the embeddings obviously are equal, both in terms of loss and any distance or angle-based metrics on the sphere. However, for visualization on a 2D map, such as a Mercator projection, which is a conformal map of the sphere, points close to the equator of this map show much less distortion in terms of distances compared to points close to the pole. This can be seen in for example maps of planet earth commonly used in an atlas or most other print media, where the arctis or antarctis appear extremely stretched—or overly large—compared to their actual size relative to e.g. Europe. For 2D visualizations of any MERCAT embedding $Y$, we hence use a rotation that puts as many points as possible close to the equator, thus avoiding as much "stretching" as possible. To this end, we compute a simple grid of rotation angles $\alpha \in [-\pi/2, \pi/2], \beta \in [0, \pi]$ with a granularity of 40 (i.e., grid values in steps of $\pi/40$) for rotation matrix $R_{\alpha,\beta} = R_\alpha R_\beta$, with

$$R_\alpha = \begin{pmatrix} \cos(\alpha) & 0 & -\sin(\alpha) \\ 0 & 1 & 0 \\ \sin(\alpha) & 0 & \cos(\alpha) \end{pmatrix},$$

$$R_\beta = \begin{pmatrix} \cos(\beta) & -\sin(\beta) & 0 \\ \sin(\beta) & \cos(\beta) & 0 \\ 0 & 0 & 1 \end{pmatrix}.$$

By evaluating a simple penalty based on the sum of squared latitudes across all points in the rotated embedding $Y^r = Y R_{\alpha,\beta}$, defined as $\sum_i (|\cos^{-1}(Y_i^r) - \pi/2|)^2$, we can optimize for a equator-favoring rotation for visualization purposes. We use this approach to generate any 2D maps of MERCAT embeddings.

## C.7 SYNTHETIC DATA RESULTS

We give visualizations of the generated embeddings for synthetic data in Fig. 10, 11, 9 and quantitative evaluation in Tab. 3, 4. All visualizations are for seed 1 of the repeated experiments, results are visually very similar across seeds, as also evident from the performance metrics.

Table 3: *Synthetic and toy data results.* We report angle preservation ($\angle$), distance preservation ($\|.\|$), neighborhood preservation ($\therefore$), and density preservation ($\odot$) between computed low-dimensional embeddings and original data on synthetic benchmarks. We report mean and standard deviation across 3 repetitions of data generation, except mammoth, smiley and circle. All numbers are rounded to two decimal places, higher is better, and **best method is in bold**, second best is underlined. "$-$" indicates that the method did not converge in 24h.

| Data | Metric | MERCAT (ours) | UMAP | tSNE | NCVis | DensMAP | hMDS |
|---|---|---|---|---|---|---|---|
| Smiley | $\angle$ | **1.0** | .10 | .25 | .14 | .26 | $NA$ |
| | $\|.\|$ | **1.0** | .11 | .39 | .22 | .37 | .99 |
| | $\therefore$ | **.85** | .79 | .84 | .75 | .83 | .80 |
| | $\odot$ | **.98** | $-.16$ | $-.32$ | .27 | .88 | .96 |
| Mammoth | $\angle$ | **.95** | .56 | .50 | .16 | .68 | $-$ |
| | $\|.\|$ | **.99** | .75 | .61 | .21 | .88 | $-$ |
| | $\therefore$ | .31 | .57 | **.65** | .54 | .60 | $-$ |
| | $\odot$ | .59 | .01 | .10 | .31 | **.73** | $-$ |
| Circle | $\angle$ | **.99** | .73 | .64 | .28 | .95 | $NA$ |
| | $\|.\|$ | **.99** | .85 | .72 | .44 | .96 | **.99** |
| | $\therefore$ | .90 | .83 | .90 | .77 | .90 | **.95** |
| | $\odot$ | .77 | .10 | .47 | .20 | **.89** | **.89** |
| Unif5 | $\angle$ | **.67**$_{\pm.01}$ | .49$_{\pm.02}$ | .50$_{\pm.04}$ | .51$_{\pm.02}$ | .51$_{\pm.02}$ | $NA$ |
| | $\|.\|$ | **.90**$_{\pm.05}$ | .41$_{\pm.07}$ | .53$_{\pm.20}$ | .44$_{\pm.13}$ | .58$_{\pm.12}$ | **.90**$_{\pm.04}$ |
| | $\therefore$ | **.49**$_{\pm.02}$ | .36$_{\pm.01}$ | .37$_{\pm.01}$ | .37$_{\pm.01}$ | .37$_{\pm.00}$ | .47$_{\pm.01}$ |
| | $\odot$ | .22$_{\pm.02}$ | .18$_{\pm.03}$ | .59$_{\pm.03}$ | .45$_{\pm.03}$ | **.61**$_{\pm.04}$ | .09$_{\pm.11}$ |
| Gauss5 | $\angle$ | **.72**$_{\pm.00}$ | .53$_{\pm.02}$ | .50$_{\pm.00}$ | .57$_{\pm.01}$ | .49$_{\pm.04}$ | $NA$ |
| | $\|.\|$ | **.93**$_{\pm.00}$ | .66$_{\pm.00}$ | .52$_{\pm.00}$ | .69$_{\pm.01}$ | .45$_{\pm.12}$ | .89$_{\pm.00}$ |
| | $\therefore$ | **.49**$_{\pm.00}$ | .37$_{\pm.00}$ | .38$_{\pm.00}$ | .38$_{\pm.00}$ | .37$_{\pm.00}$ | .47$_{\pm.00}$ |
| | $\odot$ | .33$_{\pm.00}$ | .15$_{\pm.00}$ | .59$_{\pm.00}$ | .46$_{\pm.04}$ | **.65**$_{\pm.00}$ | .20$_{\pm.00}$ |
| Gauss10 | $\angle$ | **.61**$_{\pm.00}$ | .33$_{\pm.00}$ | .35$_{\pm.00}$ | .35$_{\pm.00}$ | .35$_{\pm.00}$ | $NA$ |
| | $\|.\|$ | **.82**$_{\pm.00}$ | .26$_{\pm.00}$ | .17$_{\pm.00}$ | .21$_{\pm.02}$ | .21$_{\pm.08}$ | **.82**$_{\pm.00}$ |
| | $\therefore$ | **.44**$_{\pm.00}$ | .37$_{\pm.00}$ | .40$_{\pm.00}$ | .40$_{\pm.00}$ | .38$_{\pm.00}$ | .41$_{\pm.00}$ |
| | $\odot$ | .22$_{\pm.00}$ | .09$_{\pm.00}$ | .62$_{\pm.00}$ | .56$_{\pm.00}$ | **.70**$_{\pm.01}$ | .23$_{\pm.00}$ |
| Gauss5-S | $\angle$ | **.70**$_{\pm.00}$ | .53$_{\pm.00}$ | .46$_{\pm.00}$ | .50$_{\pm.00}$ | .51$_{\pm.01}$ | $NA$ |
| | $\|.\|$ | **.90**$_{\pm.00}$ | .61$_{\pm.00}$ | .38$_{\pm.00}$ | .44$_{\pm.00}$ | .52$_{\pm.27}$ | .86$_{\pm.00}$ |
| | $\therefore$ | **.36**$_{\pm.00}$ | .24$_{\pm.00}$ | .26$_{\pm.00}$ | .25$_{\pm.00}$ | .25$_{\pm.00}$ | .33$_{\pm.00}$ |
| | $\odot$ | .38$_{\pm.02}$ | $-.06$$_{\pm.0}$ | .23$_{\pm.00}$ | .33$_{\pm.05}$ | **.57**$_{\pm.01}$ | .27$_{\pm.00}$ |
| Gauss5-D | $\angle$ | **.69**$_{\pm.00}$ | .49$_{\pm.00}$ | .49$_{\pm.00}$ | .57$_{\pm.01}$ | .50$_{\pm.01}$ | $NA$ |
| | $\|.\|$ | **.88**$_{\pm.00}$ | .49$_{\pm.00}$ | .56$_{\pm.00}$ | .76$_{\pm.01}$ | .59$_{\pm.02}$ | .85$_{\pm.00}$ |
| | $\therefore$ | **.51**$_{\pm.00}$ | .36$_{\pm.00}$ | .38$_{\pm.00}$ | .37$_{\pm.00}$ | .36$_{\pm.01}$ | .41$_{\pm.00}$ |
| | $\odot$ | .60$_{\pm.00}$ | $-.15$$_{\pm.0}$ | .06$_{\pm.00}$ | .05$_{\pm.03}$ | **.74**$_{\pm.01}$ | .57$_{\pm.00}$ |

Table 4: *Synthetic and toy data results contd.* We report angle preservation ($\angle$), distance preservation ($||.||$), neighborhood preservation ($\therefore$), and density preservation ($\odot$) between computed low-dimensional embeddings and original data on synthetic benchmarks. We report mean and standard deviation across 3 repetitions of data generation, except mammoth smiley and circle. All numbers are rounded to two decimal places, higher is better. "$-$" indicates that the method did not converge in 24h. C-Isomap did not terminate for some data due to singularity issues during an internal matrix decomposition step. MDS in 2D corresponds to the original data ($*$)

| Data | Metric | LAPEIGMAP | DIFFMAP | SPHEREMAP | C-ISOMAP | NMMDS |
|---|---|---|---|---|---|---|
| Smiley | $\angle$ | .57 | 1.0 | .06 | .73 | $*$ |
| | $||.||$ | .60 | 1.0 | .12 | .80 | $*$ |
| | $\therefore$ | .56 | .99 | .41 | .73 | $*$ |
| | $\odot$ | .76 | 1.0 | $-.03$ | .53 | $*$ |
| Mammoth | $\angle$ | .62 | .60 | .05 | $-$ | $-$ |
| | $||.||$ | .79 | .74 | .09 | $-$ | $-$ |
| | $\therefore$ | .19 | .30 | .22 | $-$ | $-$ |
| | $\odot$ | $-.12$ | .52 | .07 | $-$ | $-$ |
| Circle | $\angle$ | .99 | 1.0 | .13 | .99 | $*$ |
| | $||.||$ | .99 | 1.0 | .23 | .99 | $*$ |
| | $\therefore$ | .93 | .99 | .32 | .97 | $*$ |
| | $\odot$ | .69 | .99 | .15 | .89 | $*$ |
| Unif5 | $\angle$ | $.37_{\pm.02}$ | $.64_{\pm.03}$ | $.34_{\pm.01}$ | $NA$ | $.65_{\pm.01}$ |
| | $||.||$ | $.51_{\pm.01}$ | $.89_{\pm.01}$ | $.46_{\pm.01}$ | $NA$ | $.90_{\pm.07}$ |
| | $\therefore$ | $.34_{\pm.02}$ | $.44_{\pm.03}$ | $.35_{\pm.00}$ | $NA$ | $.50_{\pm.02}$ |
| | $\odot$ | $.16_{\pm.05}$ | $.15_{\pm.05}$ | $.12_{\pm.04}$ | $NA$ | $.19_{\pm.03}$ |
| Gauss5 | $\angle$ | $.57_{\pm.02}$ | $.64_{\pm.00}$ | $.36_{\pm.01}$ | $NA$ | $.71_{\pm.00}$ |
| | $||.||$ | $.68_{\pm.03}$ | $.81_{\pm.01}$ | $.49_{\pm.08}$ | $NA$ | $.93_{\pm.00}$ |
| | $\therefore$ | $.34_{\pm.01}$ | $.44_{\pm.01}$ | $.36_{\pm.01}$ | $NA$ | $.51_{\pm.00}$ |
| | $\odot$ | $.00_{\pm.08}$ | $.25_{\pm.05}$ | $.04_{\pm.10}$ | $NA$ | $.19_{\pm.00}$ |
| Gauss10 | $\angle$ | $.48_{\pm.04}$ | $.55_{\pm.01}$ | $.27_{\pm.01}$ | $.55_{\pm.00}$ | $.57_{\pm.00}$ |
| | $||.||$ | $.68_{\pm.00}$ | $.71_{\pm.00}$ | $.25_{\pm.02}$ | $.76_{\pm.00}$ | $.85_{\pm.00}$ |
| | $\therefore$ | $.37_{\pm.00}$ | $.44_{\pm.00}$ | $.37_{\pm.00}$ | $.39_{\pm.00}$ | $.45_{\pm.00}$ |
| | $\odot$ | $.12_{\pm.01}$ | $.09_{\pm.02}$ | $.04_{\pm.06}$ | $-.03_{\pm.00}$ | $.10_{\pm.00}$ |
| Gauss5-S | $\angle$ | $.58_{\pm.01}$ | $.57_{\pm.00}$ | $.37_{\pm.01}$ | $NA$ | $.64_{\pm.00}$ |
| | $||.||$ | $.64_{\pm.01}$ | $.65_{\pm.02}$ | $.59_{\pm.05}$ | $NA$ | $.90_{\pm.00}$ |
| | $\therefore$ | $.18_{\pm.03}$ | $.32_{\pm.01}$ | $.24_{\pm.00}$ | $NA$ | $.35_{\pm.00}$ |
| | $\odot$ | $.13_{\pm.07}$ | $.41_{\pm.02}$ | $-.09_{\pm.04}$ | $NA$ | $.29_{\pm.00}$ |
| Gauss5-D | $\angle$ | $.21_{\pm.03}$ | $.58_{\pm.01}$ | $.37_{\pm.03}$ | $NA$ | $.64_{\pm.00}$ |
| | $||.||$ | $.26_{\pm.04}$ | $.65_{\pm.00}$ | $.49_{\pm.14}$ | $NA$ | $.88_{\pm.00}$ |
| | $\therefore$ | $.34_{\pm.02}$ | $.46_{\pm.01}$ | $.35_{\pm.00}$ | $NA$ | $.48_{\pm.00}$ |
| | $\odot$ | $.01_{\pm.06}$ | $.64_{\pm.02}$ | $-.12_{\pm.10}$ | $NA$ | $.56_{\pm.00}$ |

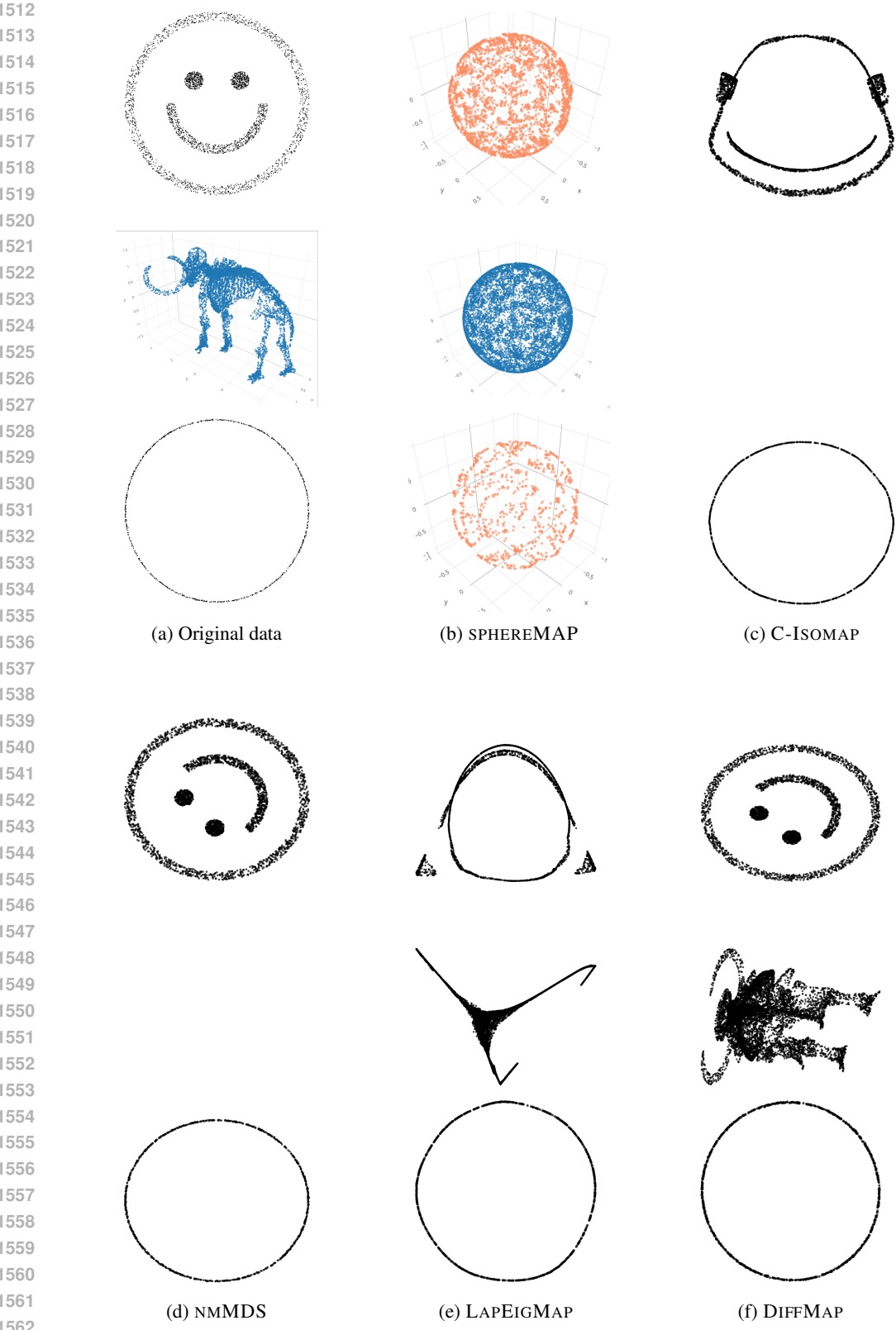

(a) Original data      (b) SPHEREMAP      (c) C-ISOMAP

(d) NMMDS      (e) LAPEIGMAP      (f) DIFFMAP

Figure 9: *Embeddings of low-dimensional examples contd.* We visualize the Smiley (top), Mammoth (middle), and Circle (bottom) data and computed embeddings. C-ISOMAP and NMMDS did not converge in reasonable time on mammoth. NMMDS in 2D corresponds to ground truth as it is equivalent to PCA in this case.

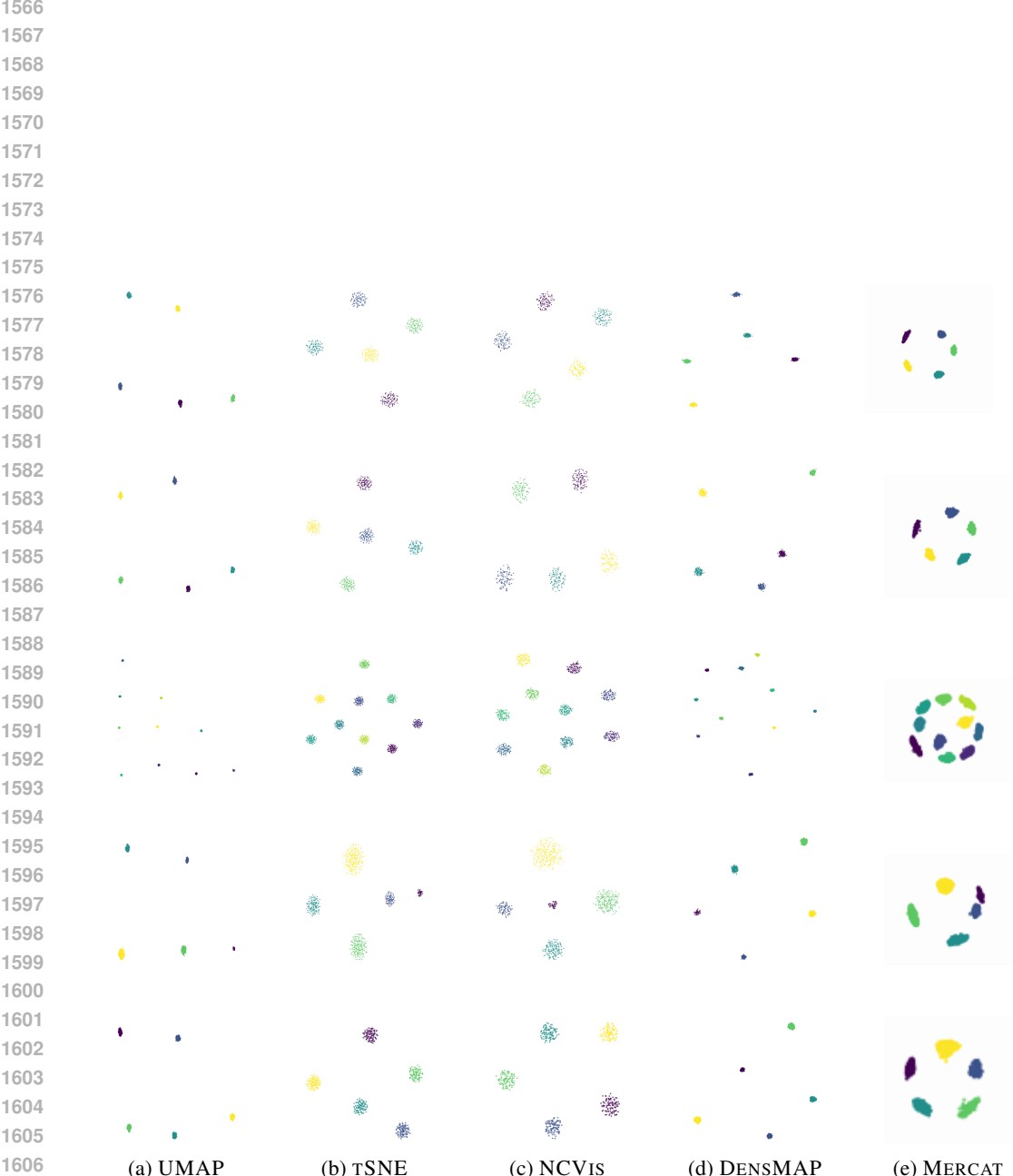

(a) UMAP      (b) TSNE      (c) NCVIS      (d) DENSMAP      (e) MERCAT

Figure 10: *Embeddings of synthetic data.* Visualizations for synthetic data sets for one random seed. From top to bottom: Unif5, Gauss5, Gauss10, Gauss5-*S*, and Gauss5-*D*. Coloring is according to cluster labels, we provide the 2D Mercator projection of MERCAT.

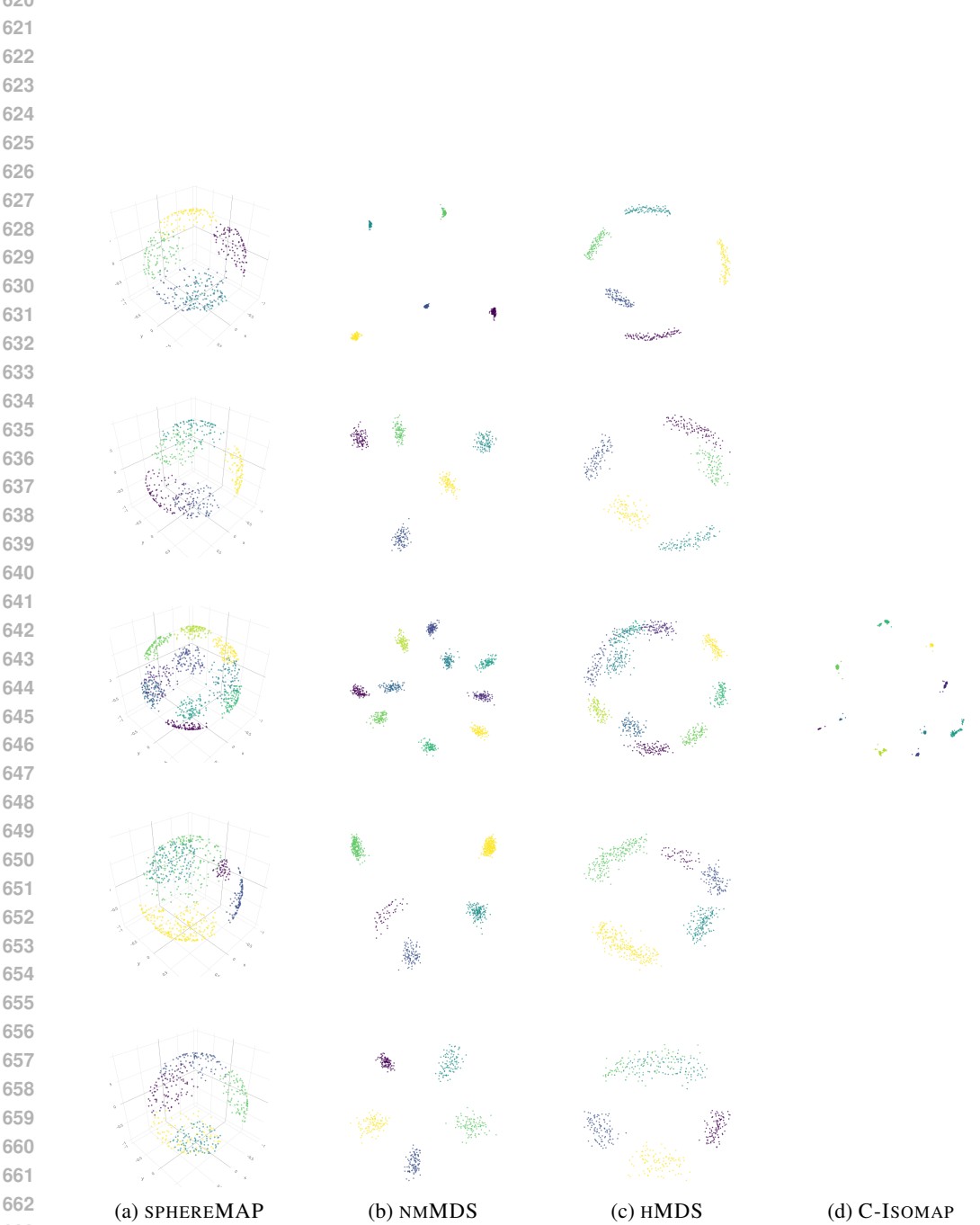

(a) SPHEREMAP  (b) NMMDS  (c) HMDS  (d) C-ISOMAP

Figure 11: *Embeddings of synthetic data contd.* Visualizations for synthetic data sets for one random seed. From top to bottom: Unif5, Gauss5, Gauss10, Gauss5-*S*, and Gauss5-*D*. Coloring is according to cluster labels. C-ISOMAP had numerical issues due to an internal matrix factorization for several datasets.

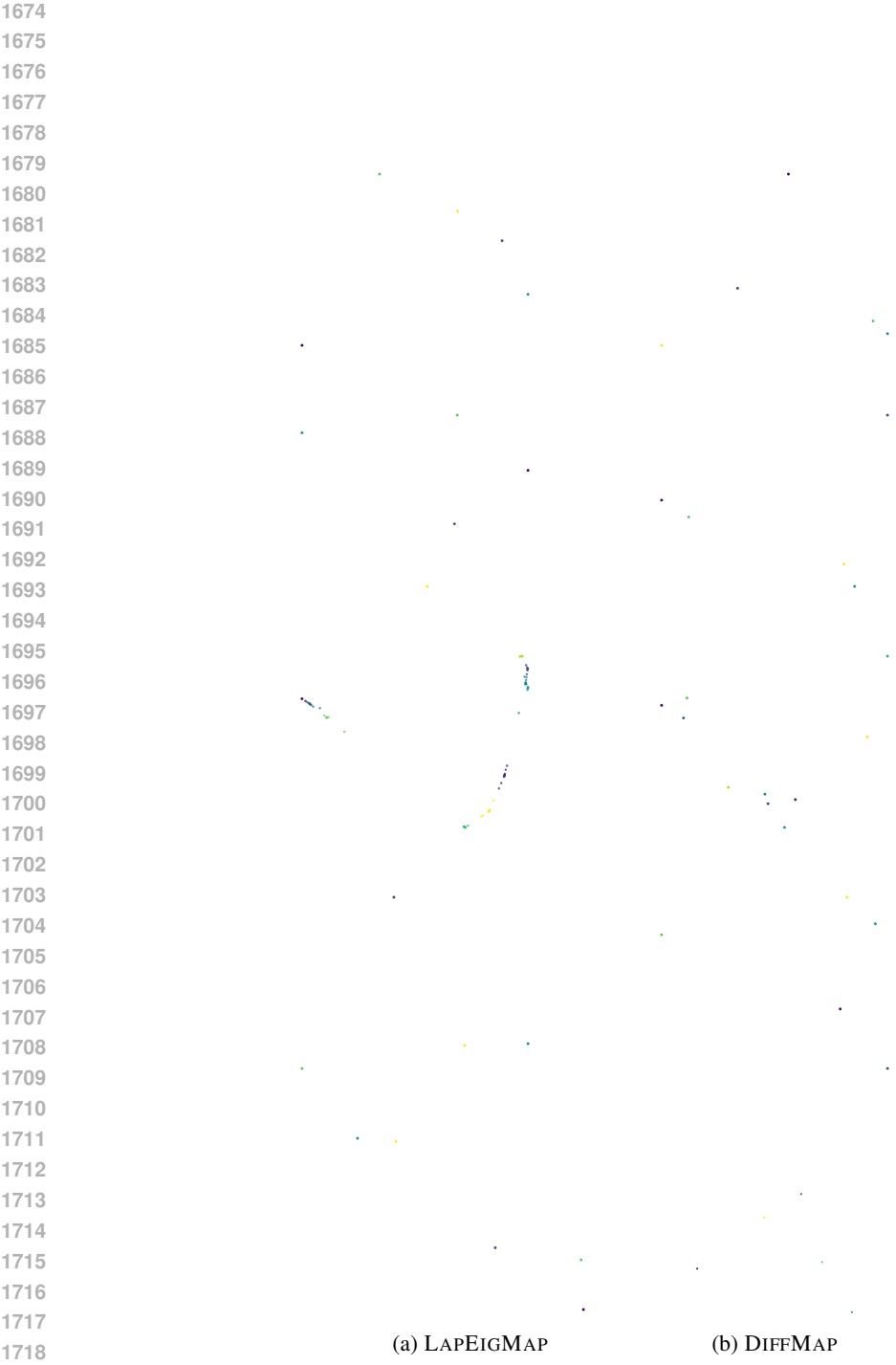

(a) LAPEIGMAP          (b) DIFFMAP

Figure 12: *Embeddings of synthetic data contd.* Visualizations for synthetic data sets for one random seed. From top to bottom: Unif5, Gauss5, Gauss10, Gauss5-*S*, and Gauss5-*D*. Coloring is according to cluster labels. Both LAPEIGMAP and DIFFMAP show the known issue of projecting points of within well separated clusters onto each other.

Table 5: *Real data results contd.* We report angle preservation ($\angle$), distance preservation ($\|.\|$), neighborhood preservation ($\therefore$), and density preservation ($\odot$) between computed low-dimensional embeddings and original data. All numbers are rounded to two decimal places, higher is better, and **best method in bold**, second best underlined, "$-$" indicates method did not terminate within 24h. Methods marked with "*" were numerically unstable, reported results are after removing samples with NA/NaN embedding coordinates.

| Data | Metric | SPHEREMAP | C-ISOMAP | NMMDS | LAPEIGMAP * | DIFFMAP * |
|---|---|---|---|---|---|---|
| Tab. Sap. blood | $\angle$ | .12 | – | – | – | – |
| | $\|.\|$ | .09 | – | – | – | – |
| | $\therefore$ | .02 | – | – | – | – |
| | $\odot$ | .00 | – | – | – | – |
| Murine Pancreas $n = 50$ | $\angle$ | .23 | – | – | – | – |
| | $\|.\|$ | .40 | – | – | – | – |
| | $\therefore$ | .06 | – | – | – | – |
| | $\odot$ | $-.15$ | – | – | – | – |
| Hematop. Paul et al. $n = 50$ | $\angle$ | .29 | .82 | .83 | .76 | .01 |
| | $\|.\|$ | .39 | .86 | .94 | .79 | $-.03$ |
| | $\therefore$ | .26 | .27 | .27 | .17 | .01 |
| | $\odot$ | .16 | .16 | .62 | $-.04$ | $-.03$ |
| MNIST even $n = 50$ | $\angle$ | .26 | – | – | – | – |
| | $\|.\|$ | .27 | – | – | – | – |
| | $\therefore$ | .06 | – | – | – | – |
| | $\odot$ | $-.05$ | – | – | – | – |
| Cell Cycle $n = 50$ | $\angle$ | .12 | .07 | .38 | NA | .16 |
| | $\|.\|$ | .05 | $-.10$ | .56 | .07 | .29 |
| | $\therefore$ | .08 | .12 | .24 | .12 | .03 |
| | $\odot$ | .15 | $-.39$ | .36 | .03 | .07 |

Table 6: *kNN classification results.* We report $k$-NN classification performance averaged over $k \in \{5, 10, 20, 50\}$ for provided cell type and digit annotation, respectively. "$-$" indicates method did not terminate within 24h.

| Data | MERCAT | UMAP | TSNE | NCVIS | DENSMAP |
|---|---|---|---|---|---|
| Unif5 | 1.0 | 1.0 | 1.0 | 1.0 | 1.0 |
| Gauss5 | 1.0 | 1.0 | 1.0 | 1.0 | 1.0 |
| Gauss10 | 1.0 | 1.0 | 1.0 | 1.0 | 1.0 |
| Gauss5-*S* | 1.0 | 1.0 | 1.0 | 1.0 | 1.0 |
| Gauss5-*D* | 1.0 | 1.0 | 1.0 | 1.0 | 1.0 |
| Hematop. Paul et al. | .56 | .60 | .65 | .61 | .59 |
| Cell Cycle | .59 | .43 | .67 | .53 | .48 |

## C.8 VISUALIZATIONS FOR REAL WORLD DATA

We provide visualizations of the embeddings generated by all methods on real data in Fig. 15, 16, 17, 18, 19,20, 21 and give runtime estimates in Tab. 8, all methods being run on the same commodity hardware (CPU: 13th Gen. Intel Core i5-1350P, RAM: 32GB DDR5 5600MHz, OS: Debian 12). We further provide numerical results on additional methods in Tab. 5

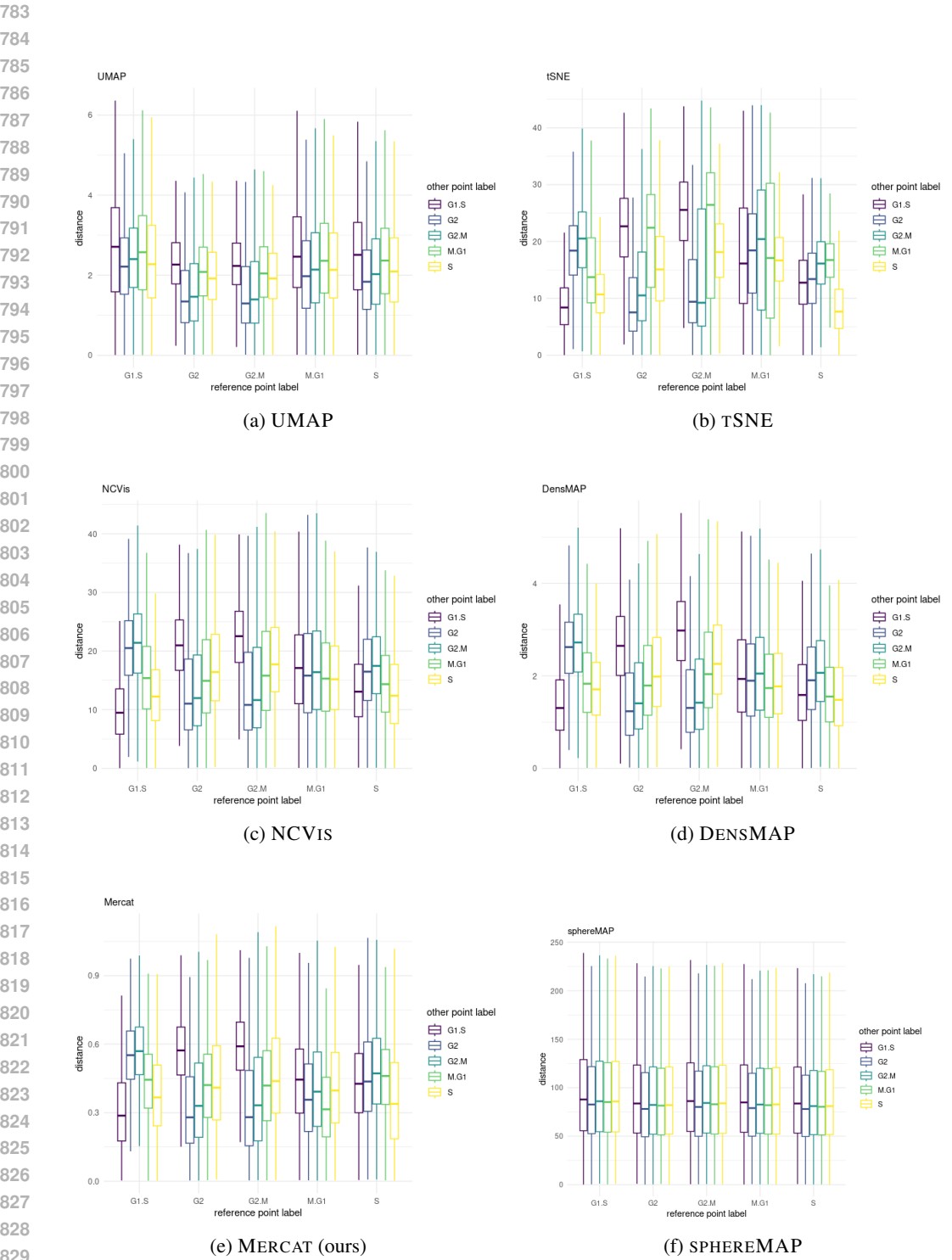

Figure 13: *Distribution of distances for cell cycle data.* We visualize the distribution of distances separated by labels, showing that methods such as TSNE do not preserve the *ordering* of labels (or clusters).

Table 7: *kNN classification results contd.* We report $k$-NN classification performance averaged over $k \in \{5, 10, 20, 50\}$ for provided cell type and digit annotation, respectively. "$-$" indicates method did not terminate within 24h.

| Data | SPHEREMAP | C-ISOMAP | NMMDS | LAPEIGMAP | DIFFMAP |
|---|---|---|---|---|---|
| Unif5 | .21 | $NA$ | 1.0 | 1.0 | 1.0 |
| Gauss5 | .20 | $NA$ | 1.0 | 1.0 | 1.0 |
| Gauss10 | .09 | 1.0 | 1.0 | 1.0 | 1.0 |
| Gauss5-$S$ | .29 | $NA$ | 1.0 | 1.0 | 1.0 |
| Gauss5-$D$ | .22 | $NA$ | 1.0 | 1.0 | 1.0 |
| Hematop. Paul et al. | .10 | .55 | .55 | .42 | .11 |
| Cell Cycle | .31 | .61 | .56 | .59 | .31 |

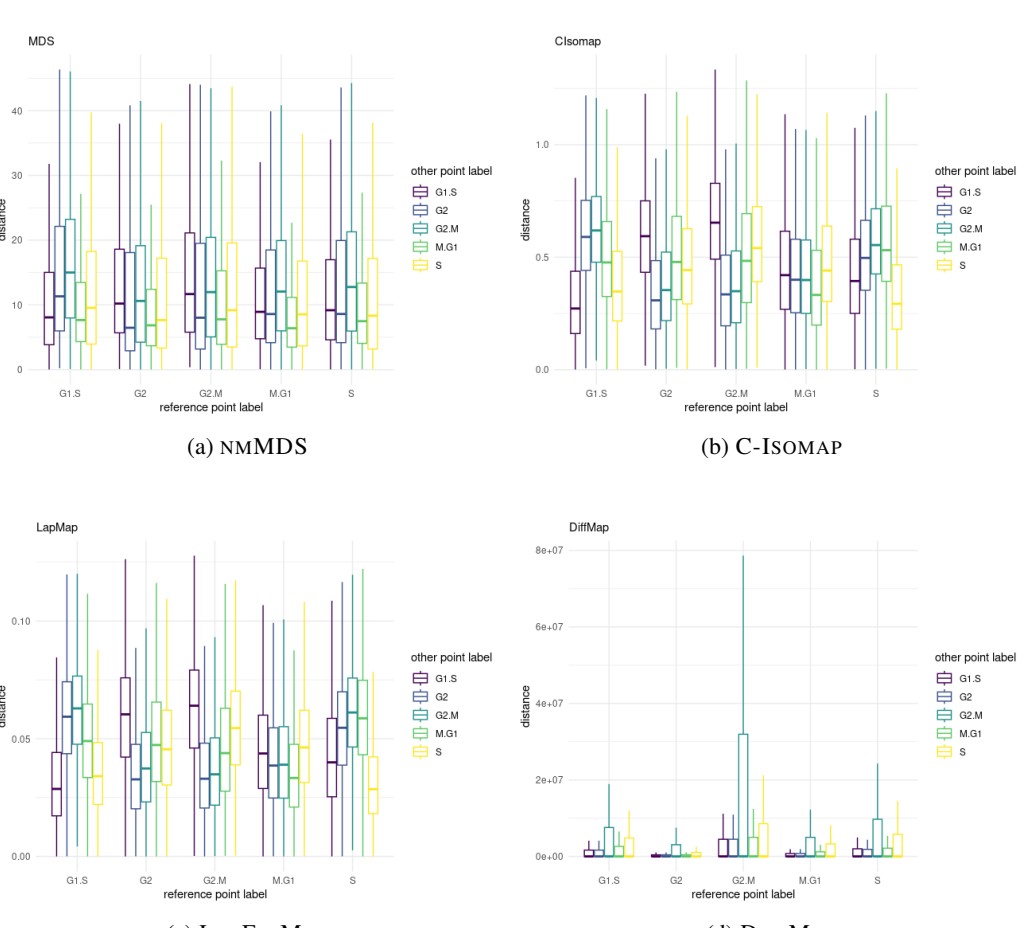

(a) NMMDS

(b) C-ISOMAP

(c) LAPEIGMAP

(d) DIFFMAP

Figure 14: *Distribution of distances for cell cycle data.* We visualize the distribution of distances separated by labels.

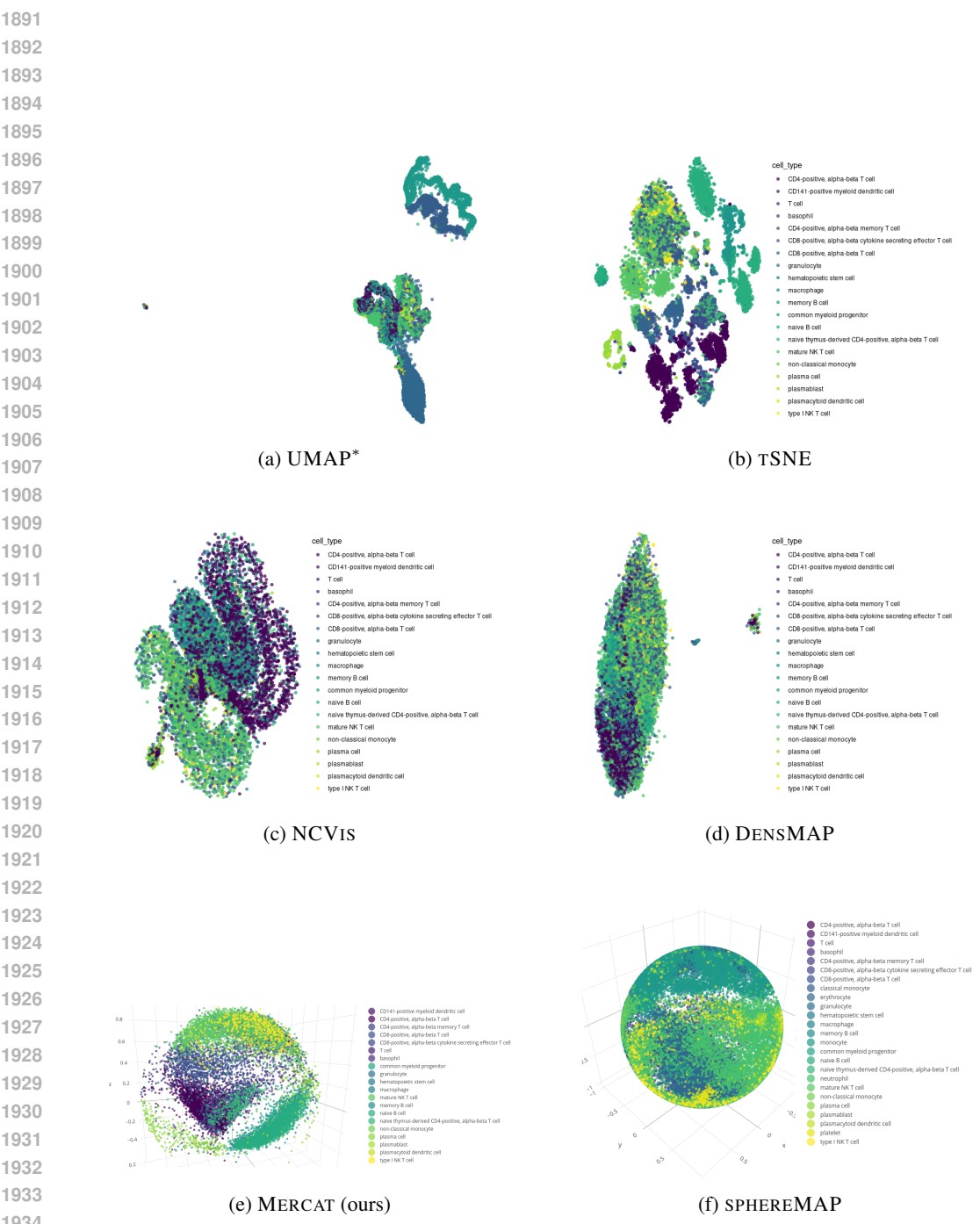

Figure 15: *Embeddings of immune related blood cells from the Tabula Sapiens project.* Coloring is according to provided cell type annotation. *UMAP did not converge to any meaningful embedding for the default parameter setting, we instead report UMAP with neighborhood parameter set to 50, which yielded good results on the Hematopoiesis data in our hyperparameter testing (see C.5)

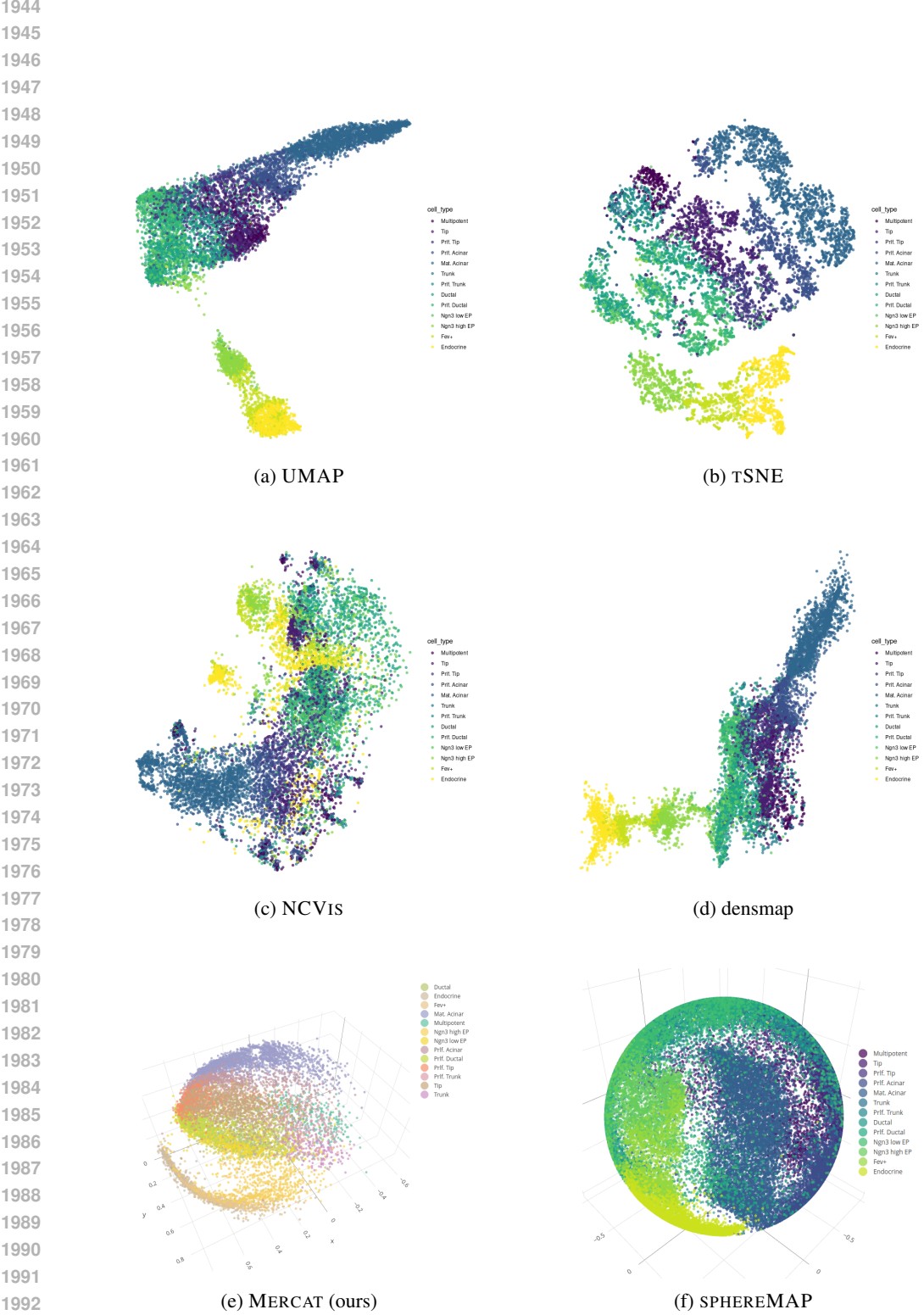

Figure 16: *Embeddings of Murine Pancreas data.* Coloring is according to provided cell annotation.

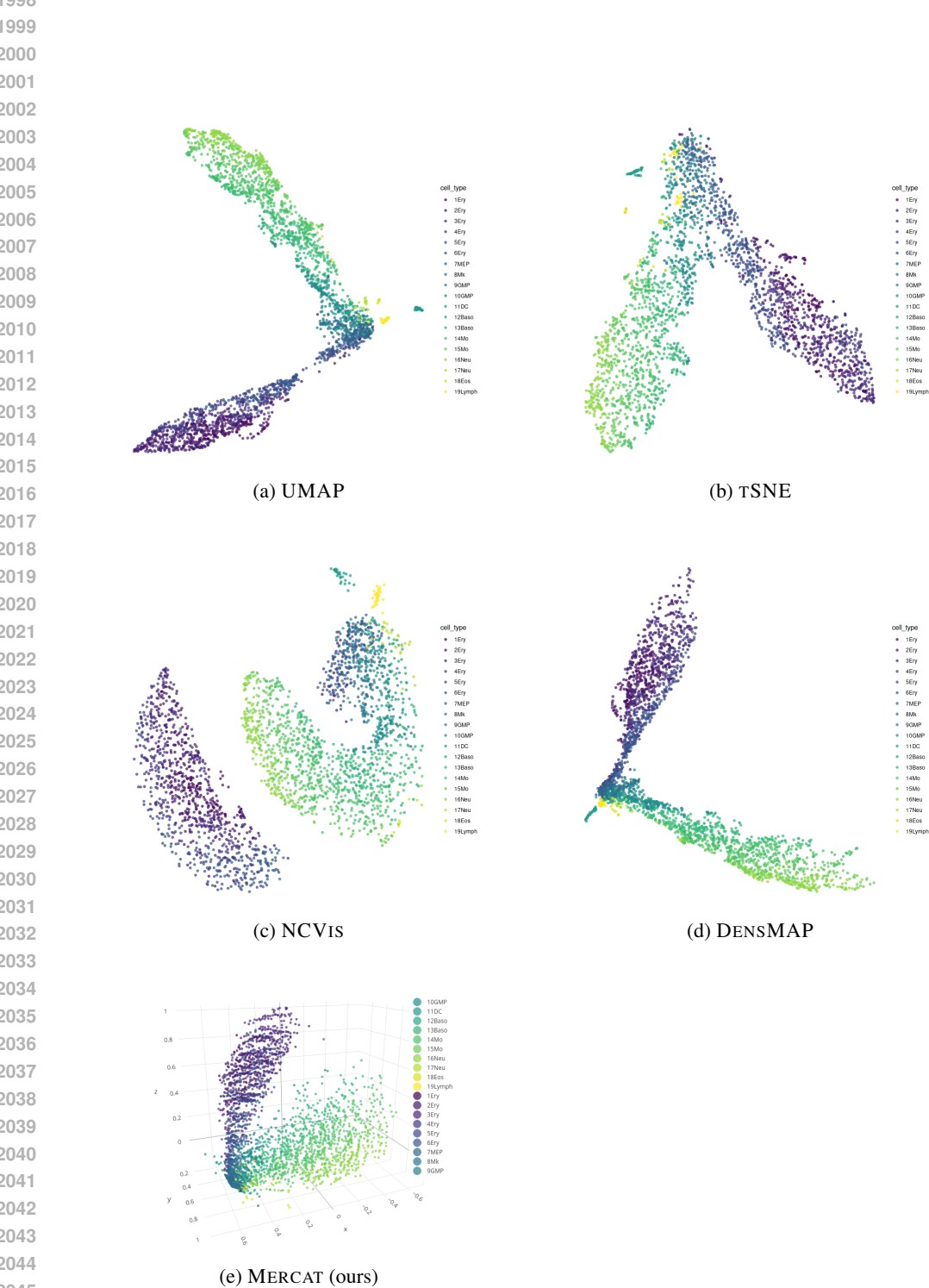

(a) UMAP

(b) TSNE

(c) NCVIS

(d) DENSMAP

(e) MERCAT (ours)

Figure 17: *Embeddings of Hematopoiesis data of Paul et al.* Coloring is according to provided cell type annotation.

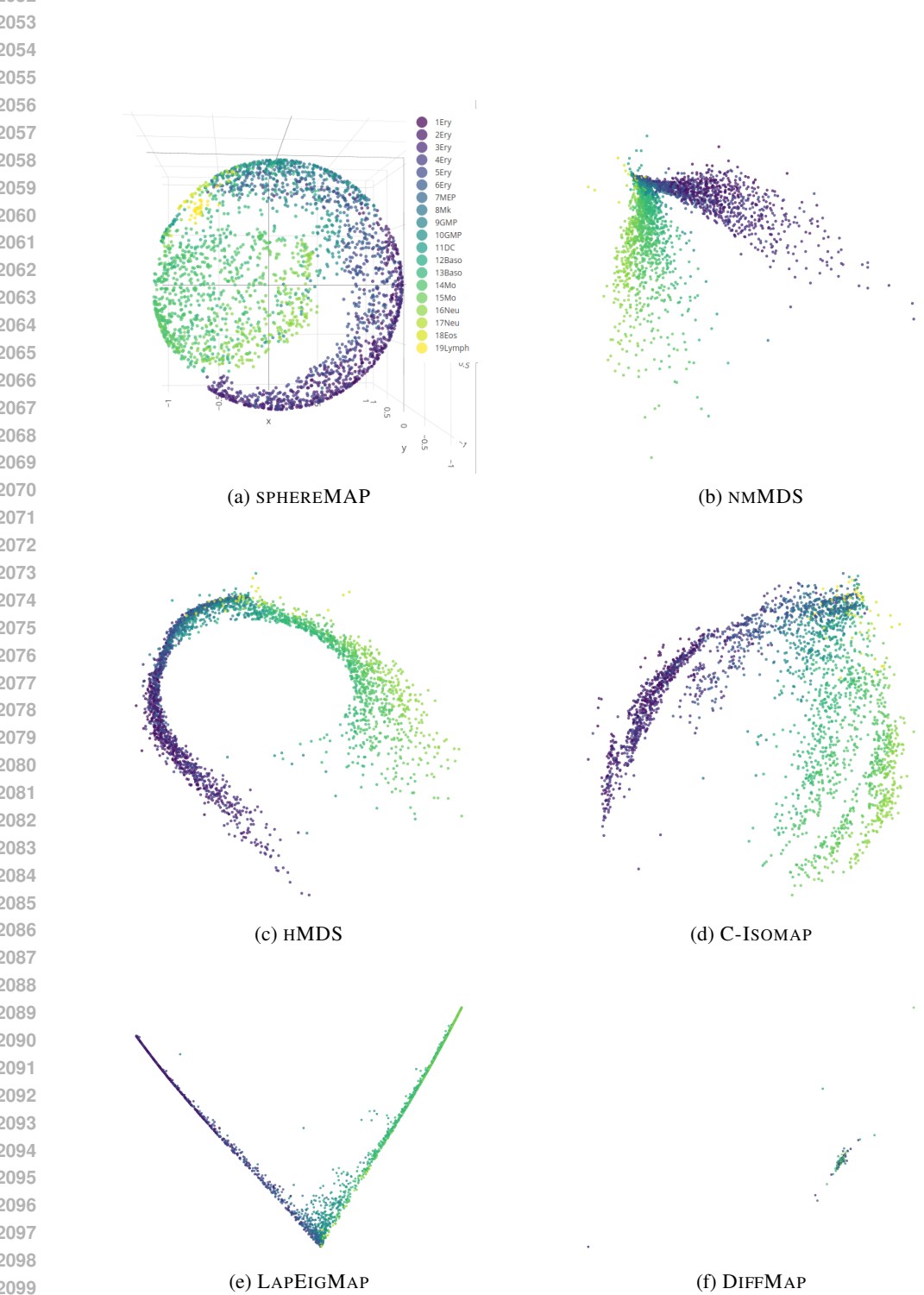

Figure 18: *Embeddings of Hematopoiesis data of Paul et al. contd* Coloring is according to provided cell type annotation.

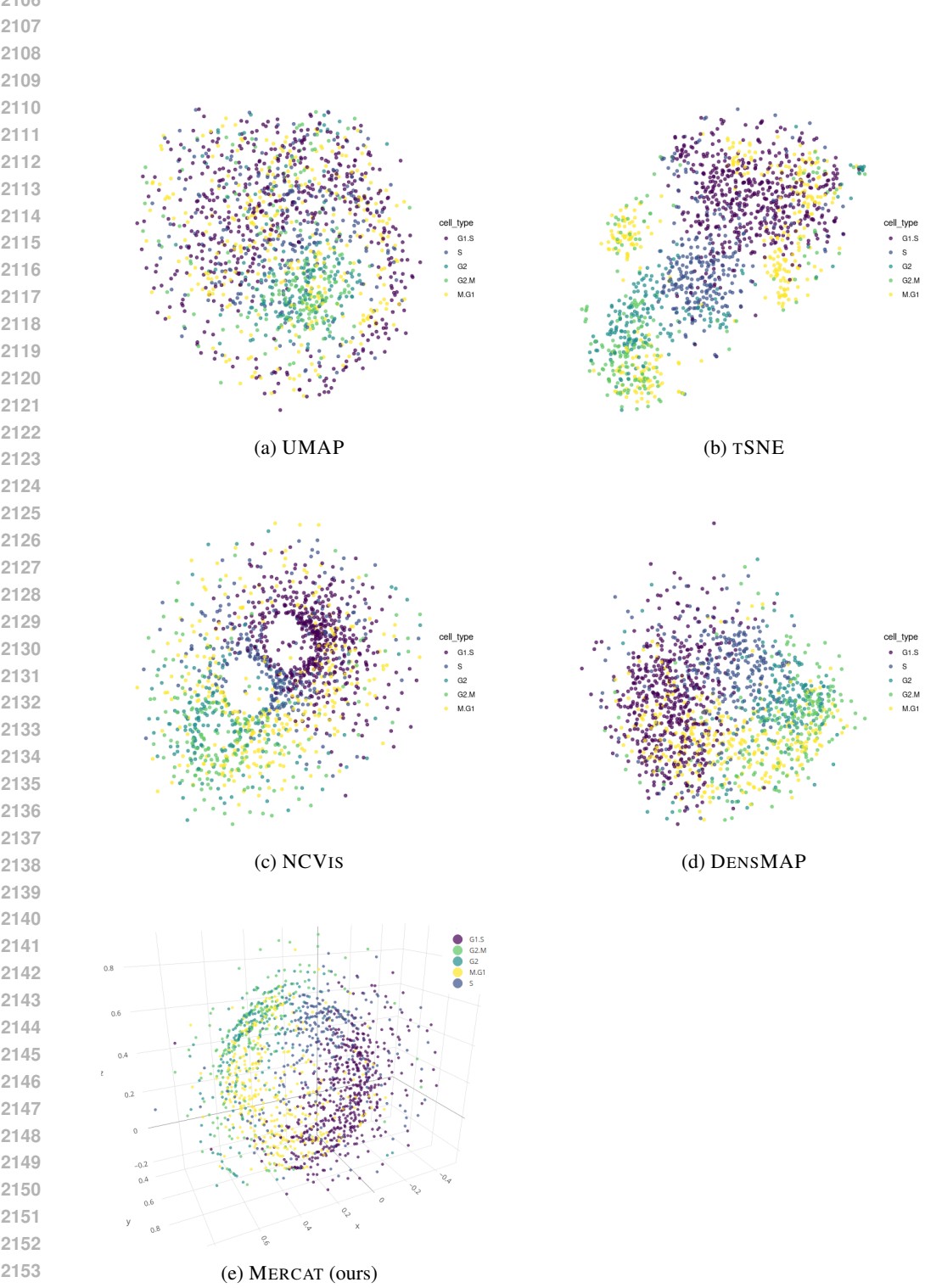

Figure 19: *Embeddings of HeLa cells across different cell cycle stages.* Coloring is according to provided cell cycle stage.

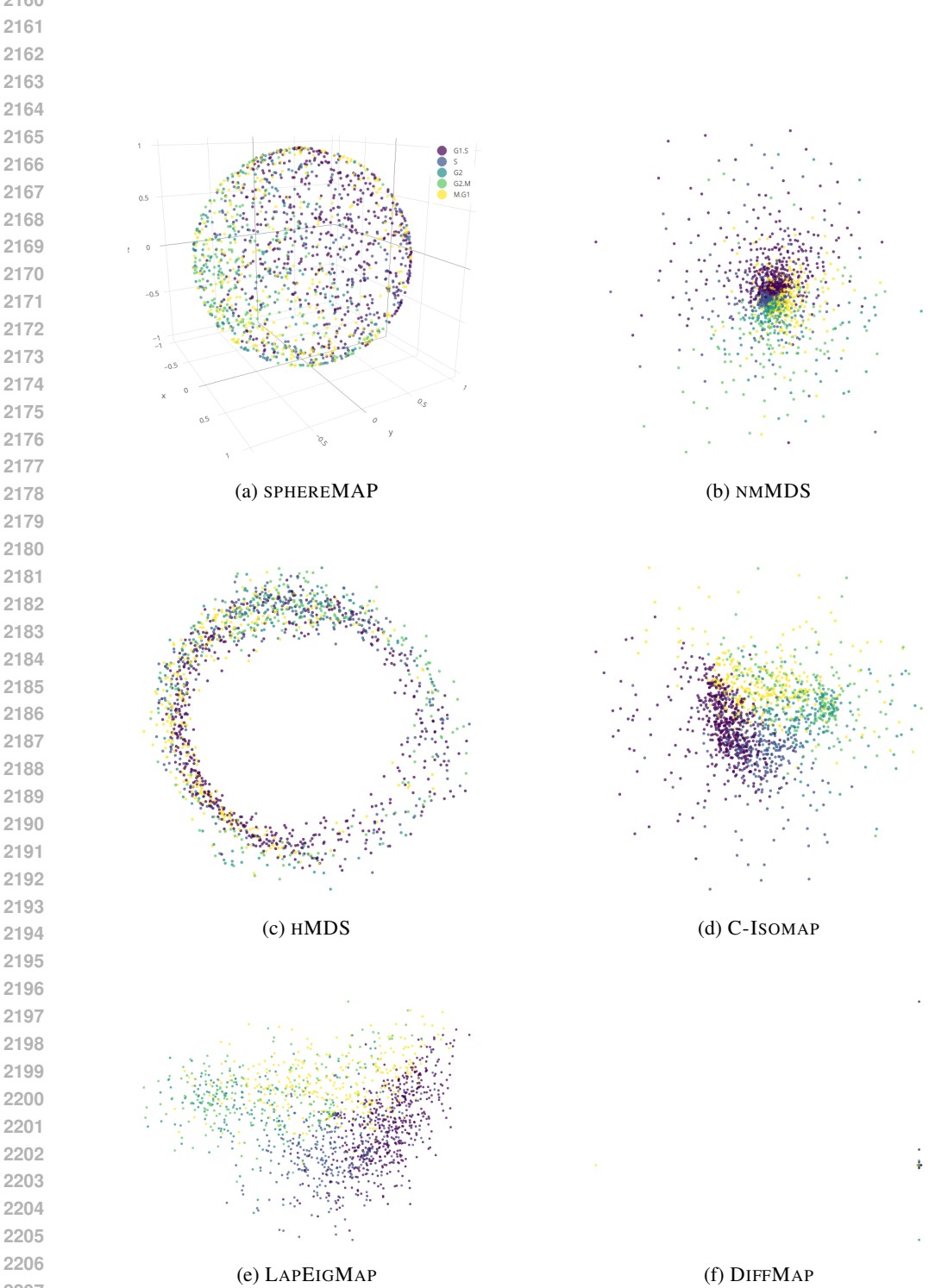

Figure 20: *Embeddings of HeLa cells across different cell cycle stages contd.* Coloring is according to provided cell cycle stage.

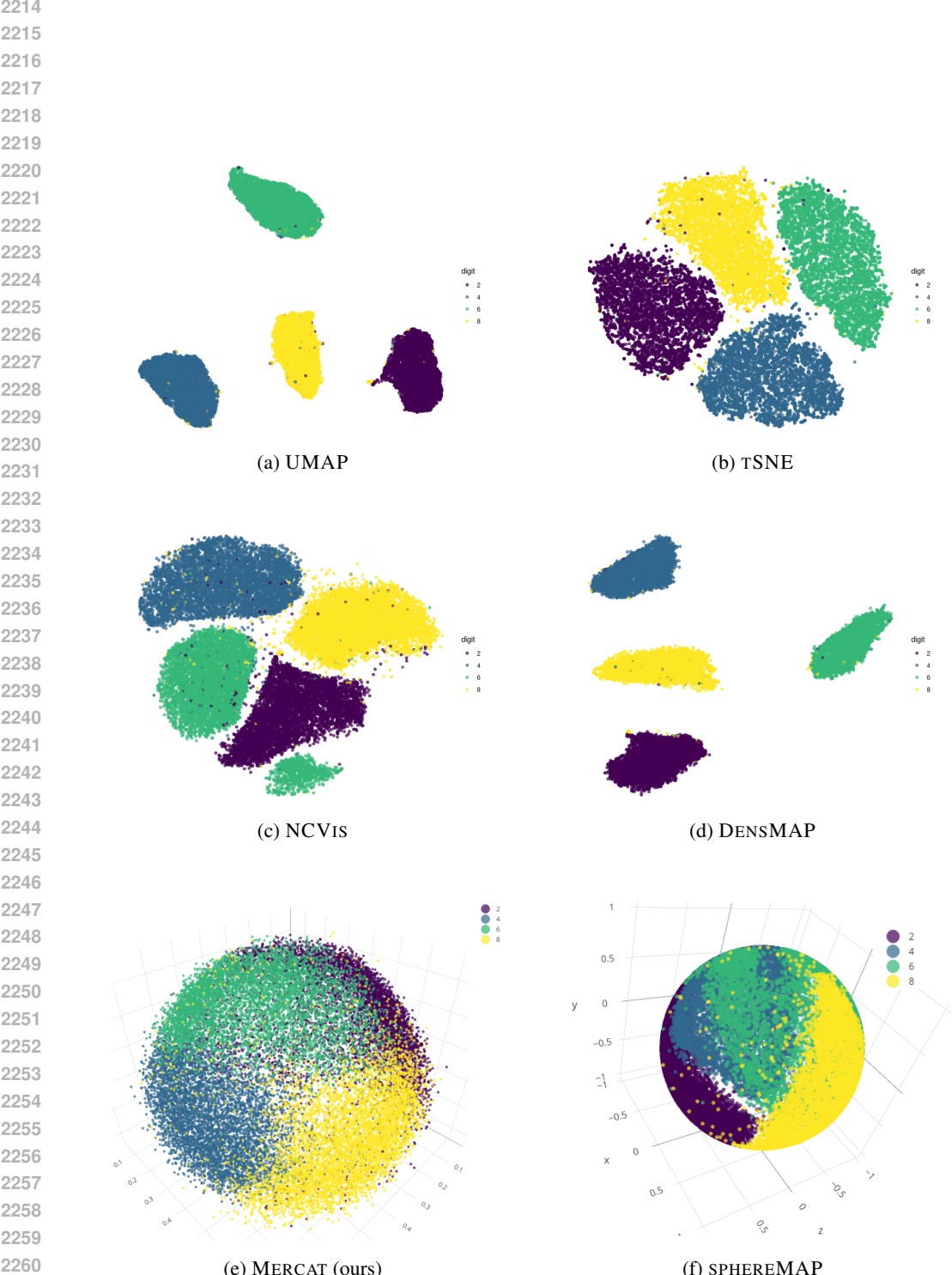

Figure 21: *Embeddings of even numbers in MNIST.* Coloring is according to digit label.

Table 8: *Runtime on real data.* We report wall clock running time for all methods. *We use a standard implementation of TSNE available in R and note that Linderman et al. (2019) proposed a faster version of TSNE. A comparison of all runtime-improvements for standard LDE approaches is out of scope of this paper, as runtime efficiency is not the prime interest. Runs marked with ∞ did not converge within 24 hours.

| Data | MERCAT (ours) | MERCAT GPU | UMAP | TSNE * | NCVIS | DENSMAP | SPHEREMAP | NMMDS | HMDS | C-ISOMAP |
|---|---|---|---|---|---|---|---|---|---|---|
| Tab. Sap. blood | 58.8m | 24.7m | 58.0m | 10.2m | 17s | 60s | 59s | ∞ | ∞ | ∞ |
| Murine | 2.2h | 20.6m | 12s | 6s | 11s | 47s | 48s | ∞ | ∞ | ∞ |
| Pancreas Hematop. | 17.7m | 18.8m | 18s | 12s | 1s | 28s | 8.1s | 1.8m | 11.7m | 23.1m |
| Paul et al. Cell Cycle | 9.1m | 6.5m | 6s | 3s | 1s | 16s | 8.3s | 1.8m | 9.9m | 22.3m |
| MNIST even | 4.2h | 3.5h | 1.8m | 1.7m | 2s | 46s | 49s | ∞ | ∞ | ∞ |

