# OpenReview forum: "Sailing in high-dimensional spaces: Low-dimensional embeddings through angle preservation"
_ICLR.cc/2025/Conference — Submitted to ICLR 2025_

### Official Review · Reviewer_vXi6 · 2024-10-21

**Soundness:** 2
**Presentation:** 2
**Contribution:** 2
**Rating:** 6
**Confidence:** 2

**Summary:**

The current generation of LDE approaches focus on reconstructing local distances between any pair of samples correctly, often outperforming traditional approaches aiming at all distances. For these approaches, global relationships are, however, usually strongly distorted, often argued to be an inherent trade-off between local and global structure learning for embeddings. The authors suggest a new perspective on LDE learning, reconstructing angles between data points. Then they show that this approach, MERCAT, yields good reconstruction across a diverse set of experiments and metrics, and preserve structures well across all scales. Compared to existing work, the proposed approach also has a simple formulation, facilitating future theoretical analysis and algorithmic improvements.

**Strengths:**

1. Originality. he authors suggest a new perspective on LDE learning, reconstructing angles between data points. Then they show that this approach, MERCAT, yields good reconstruction across a diverse set of experiments and metrics, and preserve structures well across all scales.

2. Quality and Clarity. There are no technical errors, and the presentation and writing are clear.

3. Significance. The authors show that this approach, MERCAT, yields good reconstruction across a diverse set of experiments and metrics, and preserve structures well across all scales. Compared to existing work, the proposed approach also has a simple formulation, facilitating future theoretical analysis and algorithmic improvements.

**Weaknesses:**

I am absolutely not in this field and the comments from me are not relatively professional. The comments in the following are just raised from the presentation or organization.

1. In the abstract part, the authors just state that their approach also has a simple formulation, facilitating future theoretical analysis and algorithmic improvements compared to existing work. I think the authors can add the specific values of performance gains here to better show the effectiveness of the proposed method.

2. The authors report the angle preservation, distance preservation, neighborhood preservation and density preservation between computed low-dimensional embeddings and original data. However, the authors use "-" to indicate method do not terminate within 24h. As far as I know, some experiments still report the final results that do not terminate within 24h. I think the authors can give more detailed explanations here.

3. For the real data results shown in Table 1, the authors should add more recent methods for comparison in the experiment to better show the effectiveness of the proposed method.

4. In the related work part, the authors just discuss three methods proposed after 2023, then the authors can add more recent related work for discussion in this part.

**Questions:**

1. Why choose these parts, i.e., Smiley, Mammonth, and Circle in the embeddings of low-dimensional examples？The related reasons can be given here.

2. Why choose PCA instead of the others as the manner of reduction for robustness, whihc is shown in Algorithm 1.

**Details Of Ethics Concerns:**

I have no details of ethics concerns for this work.

---

> ### Comment · Reviewer_vXi6 · 2024-11-26
>
> The authors do not give the response for this period and further discussion is not available.

---

> > ### Author Response · Authors · 2024-11-26
> > **Rebuttal timing**
> >
> > Dear reviewer,
> >
> > we thank you and the other reviewers for their questions and we are working hard to address the raised concerns.
> > In particular, we are running a large set of additional experiments, which do take time. We anticipate to upload the rebuttal and answers soon.
> >
> > Please note that the last upload time for the rebuttal is November 27th at 11:59pm AoE (i.e., in 2 days) and the discussion period, which got extended yesterday, lasts until **Dec 2nd**.
> >
> > We are looking forward to the discussion.

---

> > ### Author Response · Authors · 2024-11-27
> > **Answer to review**
> >
> > First of all, we thank the reviewer for their honesty in regard of their domain expertise and appreciate their effort to provide a structured review nonetheless.
> >
> > W1 **Performance gains in abstract**
> > Thank you for this suggestion. We adapted the abstract to mention the performance gain.
> >
> > W2 **Comparison to slow/non-converging methods**
> > Unfortunately, the considered implementations do not  offer an “online” prediction, where results could have been obtained when a run is not finished or aborted. Given the scope of the experiments, which span by now 11 methods on 13 different, partially large-scale datasets and with repetitions on synthetic data, we could unfortunately not wait more than 24 hours. This would also be impractical in a real application, and by the trends of scaling with number of sample we observed on smaller data, it would likely take weeks for specific methods to finish on larger real-world data.
> >
> > W3+4 **2023 and later in related work**
> > We compared to the state-of-the-art of the field, with tSNE and UMAP still being the dominating methods in both theoretical studies as well as applications. Since they are such a strong baseline, little progress in terms of *novel* approaches have been achieved. Newer approaches (such as densMAP, which we compare to or the weaker dtSNE), work on improving either small components of the methodology or on transferring it to different applications, such as graphs. Hence, we compare to the **strongest, most recent** work.
> > We are happy to add more work for this general setting we consider here if exists, and would appreciate it if the reviewer could point us to such recent work proposed after 2023.
> >
> >
> > Q1 **Motivation of low-dimensional data**
> > We describe the motivation behind these datasets within the paragraph “Low-dimensional data” and to summarize, these cases exemplify *known problems of current state-of-the-art methods*, including difficulties arranging clusters in the *correct orientation*, as well as properly *reconstructing manifolds*. They are in 2D or 3D, so we as the reader can *visually perceive* the difference between ground truth and reconstructed data. Furthermore, they reflect properties that are common for data of interest, such as genomics data, which often arranges along a manifold and has (sub)clusters which need to be correctly arranged to *draw the correct conclusions for the domain* (e.g. biology).
> >
> >
> > Q2 **Why PCA for initial dimensionality reduction**
> > We acknowledge that PCA is not the only suitable approach to reduce dimension or denoise the data and our approach is agnostic to the type of denoising. Our practical preference on PCA is mainly due to its low computational cost, interpretability, and its favorable analytical properties such as consistency and optimality under diverse statistical models.

---

### Official Review · Reviewer_QWRi · 2024-10-25

**Soundness:** 1
**Presentation:** 2
**Contribution:** 2
**Rating:** 3
**Confidence:** 4

**Summary:**

The paper presents a low-dimensional embedding method named MERCAT that preserves angular relationships to maintain both local and global data structures. Unlike traditional methods that prioritize local distances and often distort global structures, MERCAT maps high-dimensional data onto a 2D unit sphere by optimizing angles between points. This approach is claimed to achieve consistent structure preservation across scales and advantages over methods like UMAP and tSNE in retaining meaningful relationships within complex datasets.

**Strengths:**

* An alternative way for nonlinear dimensionality reduction
* Some theoretical guarantees are provided

**Weaknesses:**

* The paper is about visualization. But there are no figures that visualize any real-world datasets in the main paper. There are some visualizations in the appendix. However, the results are even worse than t-SNE and UMAP. Especially when the colors are removed in an unsupervised setting, the MERCAT results are messy and show little useful global pattern.
* The paper addresses faithful visualization. But faithfulness is not defined.
* The method is expensive O(n^3) and has to rely on subsampling data points.

**Questions:**

* PCA is used in Algorithm 1 to get \hat{X}. Much high-frequency information is lost in this step.
* Subsampling can lose much information about the manifold.
* Why do you multiply with 0.7*pi in the algorithm?
* Where are the definitions of performance metrics (angle preservation, distance preservation, neighborhood preservation, and density preservation)?
* A reference and discussion is missing. Faithfulness has already been discussed in the following paper.

J. Venna et al. Information Retrieval Perspective to Nonlinear Dimensionality Reduction for Data Visualization. In JMLR 2010.

---

> ### Author Response · Authors · 2024-11-27
> **Answer to review**
>
> W1 **Visual inspection of embedding visualizations "worse" than tSNE**
> This is exactly the unconscious bias we are aiming to address, and that has been discussed in the literature before. tSNE and UMAP are delivering results that **not properly reflect** the original data. They are pretty to look at, because they produce clusters, but they do NOT produce necessarily *meaningful* clusters, but rather artifacts of the optimization.
> To go beyond this subjective view that evaluates on what looks *pretty*, we evaluated a set of complementary **quantitative** metrics on a substantial amount of different datasets. In examples such as the cell cycle data, it is also clear from a **qualitative** perspective that the embeddings are wrong, as they do not align with the data (in this case, it is a cyclic structure which tSNE or UMAP do not recover at all).
>
> W2 **Terminology Faithfulness**
> “faithful” is used in its original semantic meaning, commonly used for example in XAI. We already give a meaning in line 39-40 in the introduction: “... LDEs have to be faithful to the original data: local structures should be perceivable, but also global relationships of these structures should be appropriately reflected”
>
> W3 **Complexity of method,**
> We properly discuss this in the text, with both theoretical as well empirical results, that the subsampling procedure is **efficient** and **does not harm performance**, bringing the computation down massively (O(n) in landau notation, faster than tSNE), which the reviewer seems to have overlooked.
>
> **Q1 PCA and high frequency components**
> We acknowledge that PCA may not be able to capture *all* the information contained in data. But for high-dimensional data, as we extensively argued in Section 3.4 backed up with literature from statistics, directly working on the noisy datasets without proper denoising may lead to severely biased outcomes. If the reviewer disagrees with that literature, we would be happy to discuss this in terms of science, with proper references backing up the made claims.
>
> Q2 **Supposed loss of information due to subsampling**
> See comment for W3. We extensively discussed the subsampling and showed that both theoretically and empirically it does not lose much information.
>
> Q3 **Clarification of PCA initialization**
> We assume you are talking about the PCA initialization, which we explain in line 226-227 and footnote 1 on page 5. In a quick summary:
> The optimization is slower around the poles when using gradient descent, as loss landscape is flat around them due to the way longitude and latitude are defined (involving cosines).
>
> Q4 **Where are definitions of performance metrics**
> In Appendix C2 as also referenced in the main paper.
>
> Q5 **Additional reference**
> Thank you for the suggested paper, we added it to our related work.

---

### Official Review · Reviewer_etwG · 2024-11-04

**Soundness:** 2
**Presentation:** 2
**Contribution:** 2
**Rating:** 3
**Confidence:** 3

**Summary:**

The authors propose a new technique that embeds a given dataset on  a 2-dimensional sphere within 3-dimensional space, while approximately preserving the angles between any triplets of points.

**Strengths:**

See questions.

**Weaknesses:**

See questions.

**Questions:**

The idea behind the paper is interesting, and the authors even try to convey theoretically how to extract relevant information from the given dataset (angles between triplets of points). However, from the main text it is hard to find clear results why angle preserving embedding is preferable over the current embedding techniques. I did not see a clear theorem or clear empirical results on real datasets that show the preferability of such embedding. In the next lines I will go over some of the problems I had with the main text-

1. General comment- From my understanding of the paper you are working on visualization problems which is more specific than the low dimensional embedding algorithms.

2. The Related work is one big paragraph of 32 lines. It would be preferable if this would have been broken into multiple paragraphs.

3. Initialization- The proposed method is initialized using the projection of the first two PC components onto half a sphere far away from the poles. It would be beneficial if the authors can discuss why is PCA a good initialization with respect to angles. For example, if the data is a 2-dimensional swiss roll that is embedded in 3-dimensional space I am not sure that PCA is preferable.

4. Theoretical - It will be beneficial if the authors will add a theorem that describes which data structures can be embedded based on angles.  For example, in manifold learning the intrinsic dimension plays a main role when considering to embed data.

5. Theoretical- It would be beneficial to a proof for for the conclusion that appears in Equation 4 (derived from equation 2 and theorem 2).

6. Experiments- Each evaluation metric is described in a few words and only in the Appendix they describe it technically. It will be beneficial to discuss them a bit.

7. Experiments- The proposed algorithm embeds the data in 3-dimensional space with some restrictions, while it compares to visualization techniques that use 2-dimensional space. Some of the compared algorithm can be used to generate a 3-dimensional embedding, it will be beneficial to compare to their 3-dimensional embedding version.

8. Experiments- It will be beneficial to add the PCA embedding in the different experiments, as the proposed method uses it as initialization. This will allow the reader to understand the adjustments made by your method to it. For example, in Fig 2. /"Low Dimensional data" experiment - the embedding of the proposed method seems to be close to the PCA embedding.

9. Experiments- I think that it will be beneficial to show the embedding of spectral algorithms such as Laplacian Eigenmaps and Diffusion maps.

10. Experiments-  Cluster data- The authors considered experiments on Cluster Data within the main text. However, the visualization and the tables with the performance measure were left out in the Appendix.  It will be beneficial if the authors add a figure/table about it in the main text.

11. Experiments- Real data- Some of these dataset have labels. It will be beneficial to see how well the different embeddings techniques embeds samples with similar labels together. I think that such visualization/ performance measures will be more beneficial than current performance measures.

---

> ### Author Response · Authors · 2024-11-27
> **Answer to review**
>
> We thank the reviewer for their feedback and address the raised each individual concern below.
>
> (1) **Term low-dimensional embedding** Most existing work on low-dimensional embeddings is indeed interested in visualization (including tSNE, UMAP, hMDS) and thus discusses the 2- or 3-dimensional case. In principle our method, as well as these other methods, also generalizes to higher-dimensional settings, where then we compute angles between geodesics on the *hyper*-sphere. While the nomenclature is standard in this literature, we see that it might lead to confusion to people from different fields (e.g., matrix factorization or similar), where low-dimensional embeddings or representation are typically of higher dimension. We add this as a discussion point.
>
> (2) **Split up related work**
> We are happy to do that.
>
> (3) **PCA initialization**
> Indeed, the PCA might yield a suboptimal embedding, since it is just a linear projection. But if it would already be good in general (especially for your manifold example), we would not need other methods such as tSNE, UMAP, or Mercat. In principle we do need something that is giving a rough ordering of the data relative to each other, so that we do not have too many repulsive forces at once that break the optimization at the start. We do still see lots of change during optimization away from the PCA initialization, and just found that it recovers a good embedding faster than starting with a random initialization, although both work. An ideal initialization would make for a great further study, but since PCA initialization yielded good results already, we decided to not further pursue this research path. We visualize the change away from the PCA initialization, also in response to your other question, in the updated Appendix Fig. 7.
>
>
> (4) **Further theoretical investigations**
> Identifying the family of data structures where angle-preserving embeddings outperform distance-preserving ones is a theoretically fascinating yet highly challenging topic. We must leave that for future work and would be grateful for any references where this was studied in the context of other contemporary methods, such as tSNE or UMAP.
>
> (5) **Proof of Eq 4**
> We have added the proof of Equation 4 in the Appendix B.
>
> (6) **Discussion on used metrics**
> Unfortunately we had to restrict ourselves in length due to the page limit, but indeed a brief discussion benefits the paper. We added this in the beginning of the Experiments section, thanks for the suggestion.
>
> (7) **2D vs 3D embeddings**
> Note that our algorithm embeds in a **2-dimensional** space. A sphere, as a mathematical object, is 2-dimensional and can be described by two variables, longitude and latitude. Hence, a comparison to 3-dimensional embeddings would be unfair. Note that we also compare to the (2D) sphere embedding by UMAP (here called SphereMap) to give a direct comparison to an approach embedding on the same mathematical object.
>
> (8) **Divergence from initial PCA embedding**
> Indeed, in the simple case of 2D input data, PCA is similar. In fact, by definition, PCA yields a perfect reconstruction of the original data in this simple case, as we map from 2D to 2D. So, us not diverging from the initial PCA in this particaular 2D case is good.
> A more interesting case is on real data with complex, high-dimensional relationships. We now provide a visualization of optimization over time of our method on single cell data in App. Fig. 7, starting with t=0 (the original PCA). As mentioned also in the answer to question (3), our embedding diverges significantly from the original PCA.
>
>
> (9) **Comparison to Laplacian Eigenmaps and Diffusion Maps**
> To address these concerns, we added additional experiments comparing to Laplacian Eigenmaps and Diffusion maps on the low-dimensional, synthetic, and smaller real world data to which these algorithms could readily scale to. We updated the tables and corresponding Figures in the Appendix. Consistent with the literature, these approaches are outperformed by newer approaches such as tSNE, UMAP, or Mercat.
>
>
> (10) **Reported results on cluster data**
> This must be a misunderstanding, Table 3 and Figure 8-10 in the Appendix of the original submission (Fig. 9-11 in updated draft) provided performance and visualization of both low-dimensional as well as cluster data. Please let us know if you still can’t see them as we would need to contact the AC to sort out whether there is an issue with the platform here.

---

> > ### Author Response · Authors · 2024-11-27
> > **Answer to review part 2**
> >
> > (11) **Label-based metric**
> >
> > First of all, we disagree that such a metric would be more beneficial, because
> >
> > - The actual data X can capture additional structure that could be more relevant, or not reflect the label at all. It is a strong assumption that the label is the one and only structure present in the data, and expecting the labels to reappear is an unwanted bias (i.e., “we want to see what we know”) that is also discussed in the literature. Quantitative metrics on the original data X are thus preferred.
> > - The labels in most practical settings, including in Genomics, are usually *hierarchical* or *ordered*, which pureness or closeness score on the labels, for example a kNN classification score, does not reflect at all. Looking at clusters alone is only a *local* metric that does not consider how clusters (or labels, if they coincide) relate to each other, which is exactly what we want to solve, as current methodology is biased towards these clusters ignoring larger context or structure.
> > - Methods such as tSNE perform good on such metrics although they create *artificial* clusters and *break up clusters of the same label*. For example, consider the embedding of tSNE for the cell cycle data, the M.G1 labeled cells are split up into 4 clusters at different ends of the embedding. While the clusters themselves are clear in terms of label, hence receive a good score on something that measures how pure clusters are, with kNN the typical choice in the embedding literature.
> >
> > To confirm these intuitions and satisfy the reviewer’s concern, we ran a kNN classification based on labels on all data points from a dataset, averaging accuracy across kNN runs of k \in {5,10,20,50}. To compare across methods in the interest of time we considered only the smaller real world data. We report the results in App. Tab. 6,7. In summary, on data with *clearly separated clusters* such as synthetic data, all methods perform comparably well (actually perfect, for most methods including Mercat). For real data, Mercat, UMAP, tSNE, and NCVIS all perform well, with tSNE leading the board. Interestingly, tSNE is worst in recovering actual biological signals in both single-cell blood data (with hierarchical structure) and yeast cell cycle data (with cyclic structure), exactly reflecting point 2 and 3. Not only does tSNE break clusters of similar label, the cyclic structure is not reflected at all. To address this further, we visualize the *closeness* of points separated by label (App. Fig. 13,14), which addresses relation of labels on a more global scale. What we can see is that the intrinsic order of labels (which is underlying the cell cycle) is not correct in case of tSNE.

---

### Meta-Review · Area_Chair_AsLo · 2024-12-19

**Metareview:**

The paper proposes an algorithm for low-dimensional embedding (LDE) that aims to preserve angles between data points. Specifically, the high-dimensional data are mapped onto the 2-dimensional unit sphere. The proposed method is illustrated by numerical experiments.

Reviewers generally appreciate that the proposed idea is interesting along with its theoretical analysis. However, there are concerns both about the formulation and numerical results (see below)

**Additional Comments On Reviewer Discussion:**

- Reviewer QWRi: multiplication by 0.6.pi and addition by 0.2 pi (line 200 in Algorithm 1) (the reviewer made the typo 0.7*pi).  The authors did not respond to this question. These multiplicative factors appear arbitrary (possibly by fine tuning) and were not explained.

-  Reviewer QWRi: quality of visualization, e.g. in comparison with t-SNE and UMAP. While this is subject to debate, I agree with the reviewer that in examples such as Figures 16 and 21, t-SNE and UMPAP provide clearly more meaningful visualizations can be employed not just for visualization but also other tasks such as clustering.

- Reviewer etwG: Proof of Eq(4): The authors added the proof for this in the revised manuscript. However, I find this unconvincing.
The $\sigma_i, i=1, \ldots, r$ are determined by the eigenvalues of the population covariance and there is no guarantee that they can become arbitrarily large. A much more detailed analysis would be needed here.

These points, among others, all contribute to the reject decision of the paper.

---

### Decision · Program_Chairs · 2025-01-22

Reject